# Achieving a Superhydrophobic, Moisture, Oil and Gas Barrier Film Using a Regenerated Cellulose–Calcium Carbonate Composite Derived from Paper Components or Waste

**Monireh Imani** [1,*], **Katarina Dimic-Misic** [1], **Mirjana Kostic** [2], **Nemanja Barac** [3], **Djordje Janackovic** [2], **Petar Uskokovic** [2], **Aleksandra Ivanovska** [3], **Johanna Lahti** [4], **Ernest Barcelo** [1,5] and **Patrick Gane** [1,2]

1   Department of Bioproducts and Biosystems, School of Chemical Engineering, Aalto University, 00076 Aalto, Finland
2   Faculty of Technology and Metallurgy, University of Belgrade, Karnegijeva 4, 11000 Belgrade, Serbia
3   Innovation Center of the Faculty of Technology and Metallurgy in Belgrade Ltd., University of Belgrade, Karnegijeva 4, 11000 Belgrade, Serbia
4   Faculty of Engineering and Natural Sciences, Paper Converting and Packaging Technology, Tampere University, P.O. Box 589, 33014 Tampere, Finland
5   Omya International AG, Baslerstrasse 42, 4665 Oftringen, Switzerland
*   Correspondence: monir.imani@aalto.fi

**Abstract:** It has been a persistent challenge to develop eco-friendly packaging cellulose film providing the required multiple barrier properties whilst simultaneously contributing to a circular economy. Typically, a cellulosic film made from nanocellulose materials presents severe limitations, such as poor water/moisture resistance and lacking water vapour barrier properties, related primarily to the hydrophilic and hygroscopic nature of cellulose. In this work, alkyl ketene dimer (AKD) and starch, both eco-friendly, non-toxic, cost-effective materials, were used to achieve barrier properties of novel cellulose–calcium carbonate composite films regenerated from paper components, including paper waste, using ionic liquid as solvent. AKD and starch were applied first into the ionic cellulose solution dope mix, and secondly, AKD alone was coated from hot aqueous suspension onto the film surface using a substrate surface precooling technique. The interactions between the AKD and cellulose film were characterised by Fourier-Transform Infrared Spectroscopy (FTIR) and X-ray Diffraction (XRD) showing the formation of a ketone ester structure between AKD and the hydroxyl groups of cellulose. The presence of calcium carbonate particles in the composite was seen to enhance the cellulose crystallinity. The initial high-water vapour and oxygen transmission rates of the untreated base films could be decreased significantly from 2.00 to 0.14 $g\ m^{-2}\ d^{-1}$, and $3.85 \times 10^2$ to $0.45 \times 10^2\ cm^3\ m^{-2}\ d^{-1}$, respectively. In addition, by applying subsequent heat treatment to the AKD coating, the water contact angle was markedly increased to reach levels of superhydrophobicity ($>150°$, and roll-off angle $< 5°$). The resistance to water absorption, grease-permeation, and tensile strength properties were ultimately improved by 41.52%, 95.33%, and 127.33%, respectively, compared with those of an untreated pure cellulose film. The resulting regenerated cellulose–calcium carbonate composite-based film and coating formulation can be considered to provide a future bio-based circular economy barrier film, for example, for the packaging, construction and agriculture industries, to complement or replace oil-based plastics.

**Keywords:** waste cellulose; regenerated cellulose; cellulose-mineral composite; barrier properties; superhydrophobic film; packaging film

## 1. Introduction

Polymers from lignocellulosic biomass-based materials have received increased attention due to the significant quantity of non-biodegradable wastes arising from petroleum-based polymeric materials that have ended up in landfill or, worse, let free to contaminate

the environment [1,2]. The development of bio-based materials has accelerated in several areas, including environmental control applications, such as $NO_x$ mitigation [3,4], in the field of medicine [5], the food industry [6], and coating and packaging [7–10]. A wide range of paper and paperboard already exists for a variety of packaging applications.

The raw materials available to develop, including biodegradable and sustainable films, polysaccharides, such as cellulose, chitosan, pectin, starch, and alginate, are extensively used mainly due to their low price and abundance in nature [11]. Hence, they are potential candidates for replacing non-biodegradable synthetic polymers, in particular for food packaging. Cellulose packaging film has already been introduced as a first step to avoid plastic use—for example, in supermarket coverings and for loose goods [12,13]. However, cellulose film manufacturing currently does not provide potential for the use of circular economy resources. It is this latter aspect that we newly target as the main potential arising from the work studied here.

Cellulose is difficult to process into a non-permeable material, since its natural resource form is found in plant fibre and is insoluble in most common solvents, which has been the key challenge for cellulose bioprocessing [14]. It was reported in 2002, for the first time, that cellulose can be directly dissolved in imidazolium based ionic liquid (IL) without any pre-treatment [15]. ILs are molten salts and, essentially, are the most promising green chemistry solvents proposed for cellulose dissolution. They generally have a low melting point ($\leq$100 °C), rendering them liquid at convenient operating temperatures.

Further attractive properties, including negligible vapour pressure, low toxicity, and high chemical and thermal stability, have reinforced interest in their use for the dissolution of lignocellulosic biomass. A most recent example adopting this approach is the formation of a novel nanocomposite together with the mineral calcium carbonate. This composite has shown strength properties similar to polyethylene and polypropylene, which were achieved by controlled recrystallisation on cooling developing a cellulose crystallite size ranging from 1.4 to 3.9 nm depending on the calcium carbonate filler content and type [16,17].

We propose that such a nanocomposite structure could be used to form a thin film and act as a case study to develop the necessary barrier properties to enable us to design a genuinely functioning replacement for fossil oil-based polymers, primarily in packaging, construction and agriculture industries. Using highly pure components would also support use in medical and antimicrobial applications in particular.

The targets identified to be reached when designing packaging film include that it should have a combination of water and grease-proof properties, and for content conservation/preservation, additional gas and vapour barrier characteristics are required to protect, for example, against oxygen and water vapour diffusion under conditions of high humidity [6,18]. Therefore, we evaluate if it is possible to modify the natural hydrophilicity and hygroscopy of cellulose whilst still in IL solution and, at the same time, modify the material properties to meet the full spectrum of application needs under exposure to the relevant environmental and application conditions.

To overcome the intrinsic hydrophilic and water vapour diffusive drawbacks of cellulosic film, many researchers have focused on improving the barrier properties using fluoropolymers, silane derivatives or waxes [19]. Siloxanes, despite their superior performance efficiency when used to hydrophobise paper and board, compared to, say, wax, are expensive and often considered harmful to the environment, particularly in respect to their persistence and toxic impact on aquatic systems [20–22]. The challenge remains to incorporate accepted regulatory food and environmental contact agents such as alkyl ketene dimer (AKD) and starch.

AKD has a large number of hydrophobic groups and is one of the most widely used neutral sizing agents in the paper-making industry, where its compatibility with cellulose fibre is readily achieved via the lactone rings of AKD reacting with the hydroxyl groups on hemicellulose to form $\beta$-keto esters resulting in long hydrophobic fatty acid chains to form a film [18,20,21].

This has led workers active in the field of nanocellulose to consider incorporating it into the manufacture of nanocellulose-based films derived from mechanical and/or enzymatic breakdown of fibre pulp or chemical oxidation of cellulose microfibres [18,23–27]. Whilst we recognise and widely reference that AKD has been used regularly to hydrophobise cellulose, it has, nonetheless, never been used in the context of incorporation during the unique dissolution of cellulose using ionic liquid, nor has it been used in the case of such a regeneration process of cellulose also incorporating filler particles.

The materials formed into films from the novel regenerated cellulose composites benefit from microparticle strengthening and the possibility to generate barriers. In addition to internal bulk hydrophobising, an additional goal was set to achieve potential robust superhydrophobicity using surface coating of the films, again adopting AKD. This was accomplished by incorporating an enhanced cooling-driven deposition of AKD from hot aqueous suspension followed by heat treatment until dry.

As reported by Kostic et al. (2022) [16], the chosen source of the cellulose here is also calcium-carbonate-filled waste paper, rendering the environmental value of manufacturing a functioning barrier film from otherwise waste material a decisive contribution to the circular economy. Whilst choosing calcium carbonate as the exemplified filler, we acknowledge that certain barrier properties in one-dimensional transmission mode, such as permeation through films, could be enhanced more effectively using materials of platy morphology and a high aspect ratio.

Previous work by Liang et al. (2013, 2019) [28,29] has illustrated the advantageous properties of sericite, for example, whilst many have studied the use of high aspect ratio kaolin, montmorillonite, talc, etc. [30] showing the specific advantages of these materials. However, we stress the importance of calcium carbonate in this context directly related to its vastly greater prevalence in the source material envisaged in our circular economy proposal—namely, recycled and waste paper and board, including the dominance as waste mineral in deinking sludge arising from the paper and board recycling process.

Our aim is to retain these otherwise waste materials inside the circular economy and prevent their disuse into land-fill, etc. Furthermore, resistance to permeation is often sought-after in three-dimensions, such as in bulk-volume construction block-like materials. In such a case, it is the probability of collision between the permeating molecules and barrier particles that is critical, and this depends on the particle number, related primarily to the ratio of particle fineness per unit weight addition. In this three-dimensional context, nano calcium carbonate is as effective as other readily available platy high aspect ratio nano materials as exemplified by the design of caps for PET bottles and in containers [31,32].

By focusing initially on copy paper derived from office waste, we illustrate the potential enhancement to establish circular economy. It also constitutes a first for the continued re-use of biomass materials not entering into recycling for reasons of incompatibility with common recycling processes. In this case, the critical issues are digital print deinking not being readily suitable for the industrially developed deinking process or the cellulose fibres having become highly degraded due to over-recycling.

The novel composite film-forming materials, derived from waste via IL dissolution, are examined here in respect to their morphology and chemical interactions leading to the desired properties of water and oil resistance, gas and water vapour barrier properties, and mechanical and thermal properties. Following the aim of this work, if the targets described in this introduction can be reached, the replacement of petroleum-based polymers in packaging and eventually other applications, such as agricultural and construction films, can, in principle, be realised.

## 2. Materials and Methods

Virgin cellulose (pre-hydrolysed Kraft pulp) with an intrinsic viscosity of 494 cm$^3$ g$^{-1}$ was obtained from Stora Enso Enocell (Specialty Cellulose, Uimaharju, Finland). Copy paper that would typically become waste office paper after printing was obtained from A4 sheets of 80 g m$^{-2}$ basis weight that contained cellulose fibres and precipitated calcium

carbonate (PCC). The copy paper fibre content consisted of ~73 $w/w$% fully bleached chemical pulp, and the scalenohedral form PCC made up the remaining ~27 $w/w$%, defined using ash content measurement.

Ionic liquid (IL) 1,5-diazabicyclo [4.3.0] non-5-enium acetate ([DBNH][OAc]) was prepared by the addition of equimolar glacial acetic acid (Merck, Darmstadt, Germany) to the superbase dibutyl maleate (DBN) (Fluorochem, Hadfield, Glossop, UK), with purities greater than 99%). The IL preparation was performed in a controlled temperature glass reactor initially set at 25 °C to limit the exothermic reaction enthalpy under continuous stirring for 1 h, and subsequently, the reaction temperature was raised to 70 °C [33].

Corn starch (342.30 g mol$^{-1}$) was purchased from Merck, Germany, and two types of alkyl ketene dimer (AKD); a solid AKD, used for direct addition into the IL-dissolved cellulose dope, obtained from Solenis (PerForm SP, Wilmington, NC, USA), and in a dispersed aqueous emulsion form for application as a coating on the produced films (10 $w/w$% AKD content) from Kemira Oy, Espoo, Finland.

## 2.1. Dope Formulation and Preparation of Regenerated Cellulose Films

Two host dopes containing 13 $w/w$% of cellulose dissolved in [DBNH][OAc] IL were formulated from the pure virgin cellulose pulp highly ground (sample V), as control, and similarly ground unprinted copy paper (sample C), containing the PCC filler to create the novel composite. The dope preparation followed the procedure described by Kostic et al., (2022) [16] (Table 1). In brief, IL was liquefied in a water bath for 1 h at 80 °C, then filtered through a metal filter with 5−6 μm absolute fineness and also maintained at the initial high temperature in an oven to remove remaining crystalline components. Cellulose was then dissolved in the IL held at 80 °C stirring continuously at 30 min$^{-1}$ (rpm) for 30 min.

**Table 1.** Composition and nomenclature of the film treatment.

| Sample Abbreviation | Dope Chemical Composition in IL | Post Film-Formed Treatment |
| --- | --- | --- |
| **V** | 13 $w/w$% of virgin cellulose | - |
| **C** | 13 $w/w$% preground Copy Paper | - |
| **CA** | 13 $w/w$% preground Copy Paper/1 $w/w$% AKD | - |
| **CAO** | 13 $w/w$% preground Copy Paper/1 $w/w$% AKD | oven-dried |
| **CA-S** | 13 $w/w$% preground Copy Paper/0.5 $w/w$% AKD + 0.5 $w/w$% starch | - |
| **CA-SO** | 13 $w/w$% preground Copy Paper/0.5 $w/w$% AKD + 0.5 $w/w$% starch | oven-dried |
| **CAO+AO** | 13 $w/w$% preground Copy Paper/1 $w/w$% AKD | oven-dried, dip in hot AKD emulsion, oven-dried |
| **CAO+PAO** | 13 $w/w$% preground Copy Paper/1 $w/w$% AKD | oven-dried, pre-cooled, dip in hot AKD emulsion, oven-dried |
| **CA-SO+PAO** | 13 $w/w$% preground Copy Paper/0.5 $w/w$% AKD + 0.5 $w/w$% starch | oven-dried, pre-cooled, dip in hot AKD emulsion, oven-dried |
| **CAO+PAO+PA** | 13 $w/w$% preground Copy Paper/1 $w/w$% AKD | oven-dried, pre-cooled, first dip in hot AKD emulsion, oven-dried, pre-cooled, second dip in hot AKD emulsion (without final oven drying) |
| **CAO+PAO+PAO** | 13 $w/w$% preground Copy Paper/1 $w/w$% AKD | oven-dried, pre-cooled, first dip in hot AKD emulsion, oven-dried, pre-cooled, second dip in hot AKD emulsion, oven-dried |
| **CA-SO+PAO+PAO** | 13 $w/w$% preground Copy Paper/0.5 $w/w$% AKD + 0.5 $w/w$% starch | oven-dried, pre-cooled, first dip in hot AKD emulsion, oven-dried, pre-cooled, second dip in hot AKD emulsion, oven-dried |

To form test barrier film samples, dopes, formulated as described above, were produced additionally containing either 1 *w/w*% of AKD or a mixture of 0.5 *w/w*% starch and 0.5 *w/w*% AKD based on the cellulose content (Figure 1a,b, i.e., 0.5 *w/w*% addition based on 100 *w/w*% cellulose (present at 13 *w/w*% in IL solution). This dose level of total 1 *w/w*% hydrophobising agent(s) on cellulose was chosen based on cost grounds, as a typical economic level used in papermaking. To ensure homogeneous mixing, the additives were thoroughly dispersed in the dope using an Ultra-Turrax high shear mixer (3000 min$^{-1}$ (rpm)) for 3 min). Before the film producing step, the dope was again placed in an oven at 80 °C for a few minutes to maintain the softened, flowable dope held above the melting point of the IL.

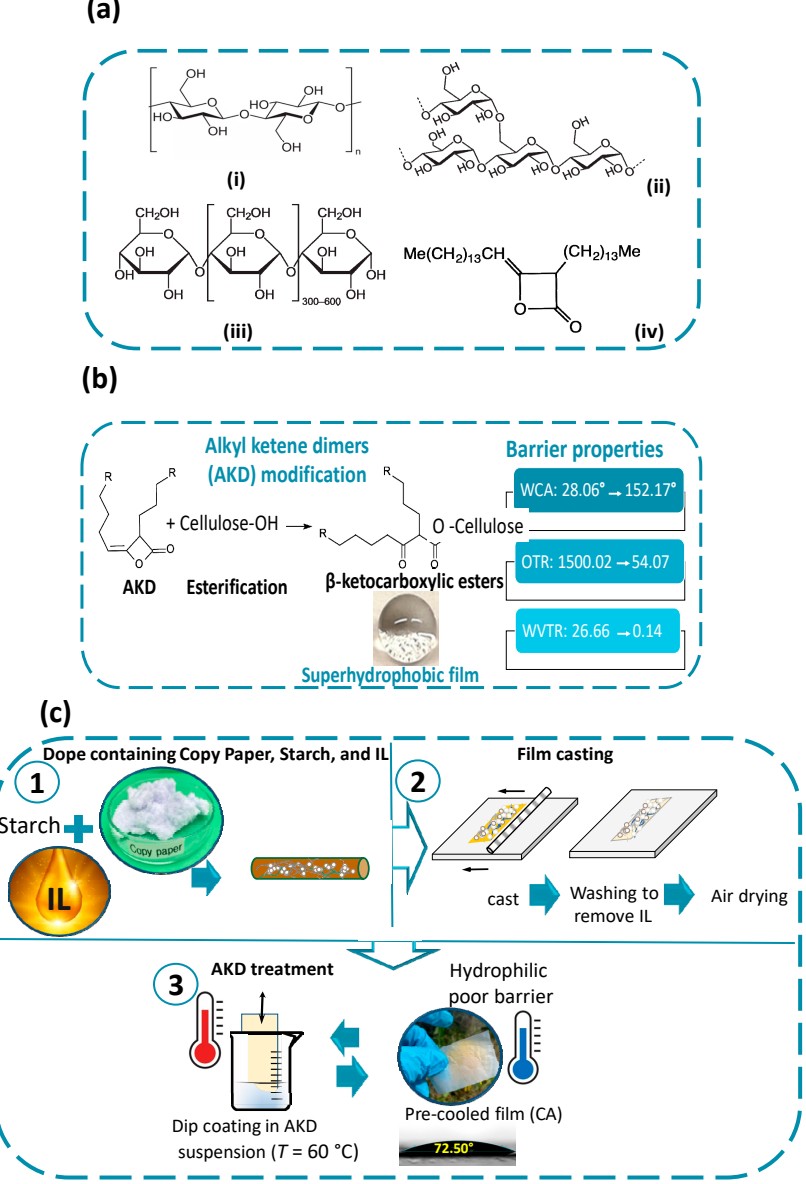

**Figure 1.** (**a**) Chemical structure of (i) cellulose, (ii) amylose, (iii) amylopectin units, and (iv) AKD, (**b**) reaction mechanism of AKD with cellulose; R representing alkyl groups, and (**c**) schematic illustration of the CA film preparation procedure, as an example, following the steps: (1) dope preparation, (2) film casting, and (3) AKD treatment.

Films were prepared using the solution-casting technique. The dope was poured onto a heated metal plate and spread to a thickness of 52 ± 2 μm using a preheated wire-wound rod on a heated laboratory coater at a speed of 3.5 m min$^{-1}$ (K control coater, model K202,

RK Print Coat Instruments Ltd., Litlington, Royston, UK). Once the surface of the wet film coagulated, the film was submerged in a bath at room temperature (a mix of 75% water and 25% ethanol) to allow for full gelation. The gel films obtained were washed with distilled water for 60 min, at 80 °C. After washing, the films were dried under planar lateral constraints at 25 °C and 50% RH (Figure 1c).

### 2.2. Surface Treatment of the Film Using Aqueous AKD Emulsion

For the preparation of highly hydrophobic, and eventually superhydrophobic film surfaces, the substrate film was coated either one or two times using an AKD emulsion suspension.

A novel approach was used to maximise emulsified AKD deposition. The dry film was first placed in a refrigerator at 5–10 °C for 60 min, then dipped directly into a preheated suspension of AKD in water (60 °C, i.e., above the melting point of AKD) with 10 *w/w*% AKD concentration. Maintaining the suspension at an elevated temperature, thus, retained the AKD in its liquid form. After a single coating, the samples were oven-dried overnight at 70 °C. Some of the single coated samples were used to apply a second AKD coating, following the same procedure except that some of the samples after the second application coat were allowed to air-dry, whilst the remainder were oven-dried as before (Figures 1c and 2).

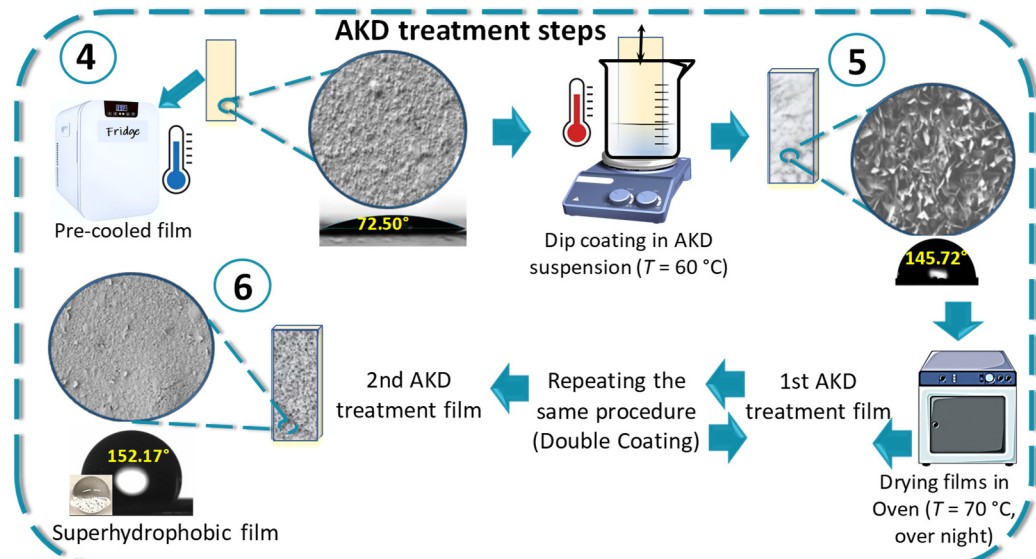

**Figure 2.** Schematic illustration of the detailed steps for AKD coating, extending the development of the surface microstructure (4) by selective multicoating (first AKD surface treatment followed by second AKD surface treatment) using aqueous AKD emulsion on pre-cooled film (5) to achieve eventual superhydrophobicity (6).

Sample labelling, reflecting the constituent and process steps, is shown in Table 1, and used throughout the remainder of this work.

### 2.3. Characterisation

2.3.1. Thickness, Density, Weight, and Porosity

Thickness, density, paper basis weight, and porosity of the films were measured according to ISO 534, ISO 536, and ISO 15901-2 standard methods, respectively (Table 2). The density of the film composites was calculated by dividing the weight in dry air by their determined volume, obtained from the thickness and surface area, respectively.

The porosity, in turn, was derived following Equation (1),

$$Porosity \ (\%) = \left(1 - \frac{\rho_{\mathrm{membrane}}}{(pph_{\mathrm{cellulose}} \times \rho_{\mathrm{cellulose}}) + (pph_{\mathrm{filler}} \times \rho_{\mathrm{filler}})/100}\right) \cdot 100 \quad (1)$$

where $\rho_{\mathrm{membrane}}$ denotes the density of membrane sheet composite, $\rho_{\mathrm{cellulose}}$ is the density of the constituent material cellulose (1460 kg m$^{-3}$), $\rho_{\mathrm{filler}}$ signifies the constituent density of the filler calcium carbonate (2710 kg m$^{-3}$), and $pph_{\mathrm{cellulose}} + pph_{\mathrm{filler}} = 100$ represents the parts per hundred by weight making up the total membrane consisting of cellulose and filler, respectively; modified from [9] Imani et al., 2019b to account for the mineral filler.

**Table 2.** The mean values (±standard deviation) of physical properties of the regenerated cellulose films.

| Sample label | Thickness/ μm | Density/ g cm$^{-3}$ | Basis Weight/ g m$^{-2}$ | Porosity/ % | Coated AKD/ g m$^{-2}$ |
|---|---|---|---|---|---|
| V | 15.75 ± 5.72 | 0.62 ± 0.01 | 29.53 ± 0.86 | 57.53 | - |
| C | 34.83 ± 5.38 | 0.77 ± 0.08 | 34.89 ± 5.83 | 32.51 | - |
| CA | 22.01 ± 8.49 | 0.67 ± 0.08 | 33.98 ± 5.68 | 41.28 | - |
| CA-S | 59.00 ± 8.02 | 0.71 ± 0.03 | 42.36 ± 2.00 | 37.77 | - |
| CAO+AO | 93.80 ± 1.10 | 0.89 ± 0.01 | 49.04 ± 1.09 | 21.99 | 15.06 |
| CAO+PAO+PAO | 216.01 ± 7.23 | 1.08 ± 0.03 | 68.88 ± 1.87 | 5.34 | 34.90 |
| CA-SO+PAO | 129.33 ± 9.40 | 0.74 ± 0.02 | 52.97 ± 3.90 | 35.14 | 10.61 |
| CA-SO+PAO+PAO | 211.21 ± 9.25 | 1.20 ± 0.07 | 69.55 ± 4.55 | 5.16 | 27.19 |

### 2.3.2. Fourier-Transform Infrared Spectroscopy (FTIR), X-ray Diffractometry (XRD), and Scanning Electron Microscopy (SEM)

All the films were dried in a vacuum oven for 48 h before microstructural and chemical analyses. The functional groups in the fabricated film were analysed by Fourier-Transform Infrared Spectroscopy using a Perkin Elmer spectrometer (single diamond ATR, USA). All spectra were collected at 25 °C over the wavenumber range of 4000–400 cm$^{-1}$ at a resolution of 4 cm$^{-1}$. Triplicates of each sample were measured.

The morphologies of the novel regenerated cellulose–calcium carbonate composite-based films were observed with an optical microscope (Carl Zeiss Axioskop 40 A pol, Göttingen, Germany) under crossed polarisers at 40× magnification. Samples with dimensions of $5 \times 5$ cm$^2$ were analysed. More detailed surface and cross section microstructure of the films was investigated using a field-emission scanning electron microscope (FESEM) (Zeiss Sigma VP, Jena, Germany) with an acceleration voltage of 20 kV. The samples were prepared by fracturing the films after cooling in liquid nitrogen to avoid structural deformation, and then sputter-coated with an iridium layer of 4 nm thickness before scanning [34] (Imani et al., 2020).

X-ray powder diffraction patterns were collected (XRD diffractometer, PANalytical X-Pert, Worcestershire, UK) using CuKα radiation emission at 45 kV and 40 mA over the range of Bragg angle $2\theta = 5$–80°. To determine the crystallinity index, $CI_{\mathrm{cell}}$, of the crystalline cellulose component, first, the total diffractogram area for the cellulose component, including the cellulose crystalline diffraction peaks and the amorphous background, was recorded.

Secondly, after subtracting the background diffuse amorphous-related data, the $CI_{\mathrm{cell}}$ was calculated by dividing the remaining cellulose diffraction peak area alone by the total area of the original cellulose component diffractogram. This procedure is considered justified in this case even though the composite material contains other components, since the other major component is highly crystalline calcium carbonate with minimal amorphous contribution, which is a constant fraction across all samples.

In parallel, the chemical additives form only a relatively small contribution mainly to the amorphous background. Such factors in the amorphous component would act minimally to reduce the $CI_{\mathrm{cell}}$, and thus the values obtained should be considered suitable at least as a comparative measure throughout the series.

### 2.3.3. Mechanical Analysis

The mechanical properties of films were evaluated using a dynamic mechanical testing system (MTS-400/M, Easte Grinstead, West Sussex, UK) with a load cell of maximum response of 200 N. Prior to mechanical testing, the films were preconditioned in a controlled atmosphere for 48 h at 50% RH and 23 °C. The specimens were cut into $5 \times 60$ mm$^2$ strips.

The initial gap length was set to 4 cm and the elongation rate to 12 mm min$^{-1}$. The results were averaged over at least six specimens.

### 2.3.4. Water Contact Angle ($\theta_{\mathrm{WCA}}$) and Roll-Off Angle ($\theta_{\mathrm{RoA}}$)

The contact angle ($\theta_{\mathrm{WCA}}$) and roll-off angle ($\theta_{\mathrm{RoA}}$) for water on the film surface were determined with a contact angle meter (Theta flex, Biolin Scientific Inc., Phoenix, AZ, USA). A water droplet (5 μL) was automatically dispensed on the film at 23 °C and 50% RH. The average of 10 separate measurements, each recording the first observation for a fresh droplet taken after contact, is reported as the recorded $\theta_{\mathrm{WCA}}$.

The value of $\theta_{\mathrm{RoA}}$ was determined by applying a similar droplet, watching to see if it remains in the position applied, and, if so, then titling the sample on the equipment stage recording the angle at which the droplet displaces under the action of gravity and rolls freely off the surface. For superhydrophobic samples, either the droplet rolls around the surface immediately or begins to roll off the surface at a tilt angle <5°.

### 2.3.5. Water Vapour Transmission Rate (WVTR), and Oxygen Transmission Rate (O$_2$TR)

The water vapour transmission rate (WVTR) was determined according to the ASTM E96/E96 M-05 standard [35] at 23 °C and 50% RH. A specially designed cup containing oven-dried silica gel was sealed with the coated papers and edge contact sealant, and the assembled cup was then conditioned for 24 h in a desiccator at the stipulated temperature of 23 °C (room temperature) maintaining the humidity at 50 ± 5% RH to achieve moisture content equilibrium. Water vapour transport by diffusion was determined by the weight gain of the silica gel-cup-film assembly before and after the 24 h exposure in the desiccator. The values are reported as g m$^{-2}$ d$^{-1}$.

Oxygen transmission rate (O$_2$TR) of the films was determined according to ASTM D3985-05 using an oxygen permeability testing apparatus (MOCON OX-TRAN 2/21, Modern Controls Inc., Minneapolis, MN, USA). Testing was conducted at 23 °C and 50% RH. The films were cut into circular shape of ~8 cm diameter (test area of ~50 cm$^2$) and sealed between an upper chamber containing oxygen (99.999% purity) and a lower chamber void of oxygen. A coulometric sensor equipped in the lower chamber measures the oxygen volume permeated through the unit area of the test film per unit time. The values are reported as cm$^3$ m$^{-2}$ d$^{-1}$.

Two samples were measured for each test.

### 2.3.6. Grease/Oil Penetration Rate

The resistance of the film to grease permeation was determined according to the standard TAPPI method T507 cm-99. In this test, we used sunflower seed oil as the test vegetable oil. From a height of about 13 mm, a drop of oil is gently released onto the surface of the test specimen. The degree of permeation is recorded as an area of oil-transmitted stain picked up by a sheet of blotting paper contacting the rear side of the sample for the given time allowed for oil to wick through the sample.

### 2.3.7. Abrasion Resistance Property

The abrasion resistance of the coated film surfaces was evaluated using the Taber Abrader (Taber Industries following the ASTM-D4060 method. The samples were conditioned at 23 °C and a 50% RH for 24 h before the evaluation. In this method, a circular coated film of 50 mm in diameter with a hole centrally located on each sample was mounted on a rotating turntable and subjected to rotary abrasion. A constant load of 1000 g on each Taber wheel (Model CS-10 Calibrase, Taber Industries, North Tonawanda, NY, USA) was used. The samples were initially weighed before abrasion. The abraded samples were subsequently dusted off after abrasion and weighed again [36]. The difference in weight was used to determine the response to abrasion.

## 3. Results and Discussion

### 3.1. Characterisation of the Composite Films

The move to applying AKD using the novel coating technique of precooling the film and dipping it into heated AKD aqueous emulsion (+PAO) compared with the conventional coating of the film starting at room temperature and dipping it into heated AKD aqueous emulsion (+AO) ensures that a greater amount of AKD is applied and that the coating layer is more homogeneous.

The densities of films were in the range of 0.62–1.20 g cm$^{-3}$; Table 2. Samples containing internal starch/AKD and the second AKD coating (CA-SO+PAO+PAO) displayed, as expected, the highest density of $1.2 \pm 0.07$ g cm$^{-3}$ ($69.55 \pm 4.55$ g m$^{-2}$ basis weight). The density was much lower for the uncoated sample (V), at only $0.62 \pm 0.01$ g cm$^{-3}$. Since the films containing filler have a major contribution to density from the filler itself versus cellulose only ($CaCO_3$ 2.71 g cm$^{-3}$ and cellulose 1.46 g cm$^{-3}$), the much lower density of the constituted film compared with either of its components suggests the presence of air voids, including surface-roughness-related voids which subsequently become filled when coated.

The thinnest film properties are dependent largely on the quality of production. Here, we stress that the films are hand-made and no claim of perfection in quality is made. The thinnest film in the series studied was $15.75 \pm 5.72$ μm, and we would not predict values for films thinner than this using the laboratory film forming method applied. However, ultimately, the largest filler particle used would define the thinnest film that could be made uniformly. The physical properties described above are summarised in Table 2.

The FTIR spectra of the regenerated cellulose/AKD films are presented in Figure 3a, in which the characteristic peaks of cellulose are observed at 1376 (the C-H bending), 1 645 (the O-H bending) and 2920 cm$^{-1}$ (the C-H stretching) [37,38]. Comparing the FTIR spectra of the AKD-containing samples with that of the characteristic cellulose only, two new bands appear at 2912 and 2848 cm$^{-1}$—namely, that of CH stretching vibrations of methyl and methylene groups and of AKD long alkyl chains, respectively.

In addition, new peaks appear at 2825, 1842, 1717, 834, and 721 cm$^{-1}$, demonstrating the formation of the ketone ester structure through the reaction between AKD and the hydroxyl groups of cellulose [25,39,40]. The relative absorbance at 3273 cm$^{-1}$ decreased with increasing AKD content, further confirming that AKD diffused and covered the cellulose chains of the cellulose/AKD films [24]. An intensive peak appearing in all samples at 1082 cm$^{-1}$ is considered to be related to C–O stretching. In comparison to the film containing virgin cellulose (V), those films made from preground copy paper display a slight displacement of the peaks around 1468, 1080, 870, and 712 cm$^{-1}$ confirming the presence of calcite, the most stable polymorph of calcium carbonate, the mineral filler constituent of the paper.

It might be questioned as to when the interaction between AKD and cellulose via the hydroxyl groups on the anhydroglucose units (AGU), referring to single sugar molecules in polymer cellulose, actually occurs. The behaviour of these hydroxyl groups has not been studied when cellulose is in solution in ionic liquid, which, being a salt melt designed to have both hydrophobic and hydrophilic moieties, could act to block the immediate in-solution interaction with AKD.

We, therefore, at this stage perhaps avoid to speculate as to when the interaction takes place, especially since an aqueous washing step is later applied to remove the ionic liquid, during which the homogeneously distributed AKD might only then undergo the interaction suggested. Considering this and the projected use within an industrial papermaking context, the use of weight fractions is adopted rather than the interaction descriptor AGU.

Figure 3b illustrates the X-ray diffractograms used to examine the crystallinity of the composite films in the regeneration process. There is an apparent significant broad peak appearing at $2\theta \approx 21.3°$. Usually, XRD diffractograms of cellulose II show three main peaks at $2\theta \approx 12.1°$, 20.1° and 21.9°. In our case, the peak at $\approx 21.9°$ appears as a shoulder on the

peak at $\approx 20.1°$ resulting in one broad peak at $2\theta \approx 21.3°$ consisting of the combination of the diffraction intensities $I_{110}$ and $I_{020}$.

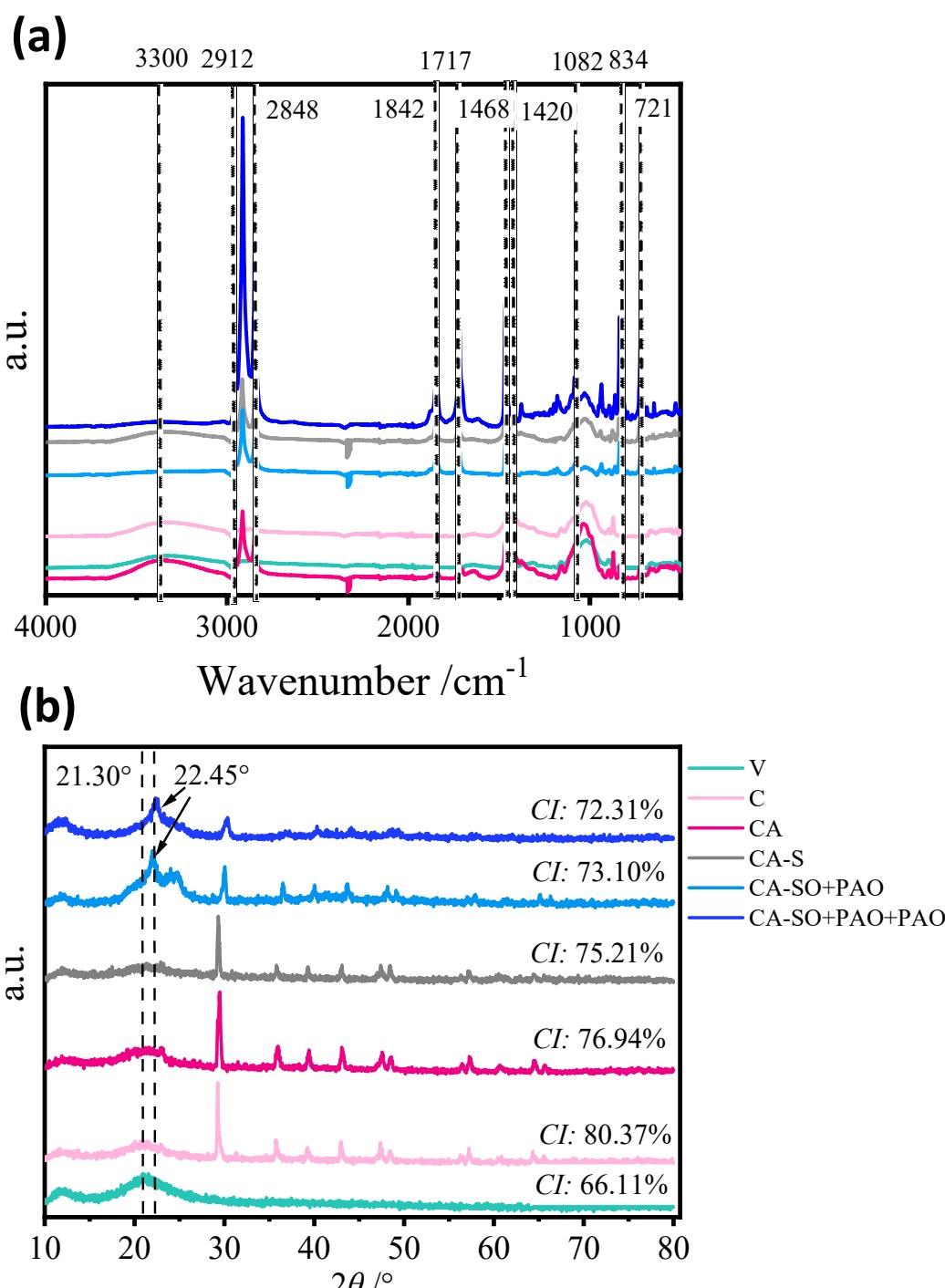

**Figure 3.** (**a**) FTIR spectra, (**b**) XRD data of the regenerated cellulose films respectively, virgin cellulose (V), ground Copy Paper (C), ground Copy Paper containing 1% internal AKD (CA), Copy Paper containing 0.5% AKD and 0.5% starch (CA-S), CA-S film single coated in AKD emulsion (CA-SO+PAO), and film with second AKD coating in emulsion (CA-SO+PAO+PAO).

The loss of the peak at $2\theta \approx 12.1°$ is expected to be due to the orientation of crystallites in the film, introduced by planar strain arising from the lateral constrainment during film drying, thus suppressing the diffraction intensity from the (1-10) crystal plane. As AKD is added into the dope (samples CA-SO+PAO and CA-SO+PAO+PAO), the cellulose peak

becomes submerged in the shoulder of the newly appearing neighbouring peak associated with AKD and its interaction with cellulose seen at $2\theta \approx 22.45°$. The cellulose crystallinity indices $CI_{cell}$, for films C and CA are 80.37% and 76.94%, respectively.

The increased crystalline nature displayed by the emerging AKD related neighbouring peak correlates strongly with the application of heat by oven drying when AKD is present either internally in the bulk film or applied afterwards to its surface. The crystalline transformation of the AKD when heated is thus more marked than the recrystallisation undergone by the cellulose during regeneration.

Due to the esterification of cellulose by AKD, abundant exposed methyl and methylene groups over the fibre surfaces provide the required hydrophobicity of the regenerated cellulosic films [38], and this might even lead to a change in cellulose-AKD interacting crystallite structure. As shall be shown later, the surface structure of the AKD coating changes during the oven drying and this can be correlated with the markedly increased crystallinity observed by XRD in the form of the related emerging peaks. In parallel, the presence of PCC filler (samples starting from copy paper C) enhances crystallinity at the higher angle structure compared with pure cellulose film V without filler ($CI_{cell}$ = 66.11%).

Similar research for mechanically and oxidatively formed nanocellulose films, showed decreased cellulose crystallinity in the case when AKD was included [37,38], demonstrating that such films containing AKD were more amorphous in structure. In the case studied here, we see only a slight decrease in $CI_{cell}$ for samples CA-SO+PAO and CA-SO+PAO+PAO (73.10% and 72.03%, respectively) in comparison with uncoated film CA-S ($CI_{cell}$ = 75.21%) though the breadth of the pure cellulose peak indicates relatively low crystallinity for the regenerated cellulose case studied here.

Precipitated calcium carbonate (PCC) presents its crystalline structure with peaks at $2\theta$ values of 29.4°, 36.00°, 39.44°, 43.20°, and 47.57°. It is seen from the XRD that PCC disperses completely in the cellulose matrix. The composite without filler (V), as is to be expected, shows no peak related to the PCC [41].

The comparison made between optical and SEM images enables the viewer to establish a statistical relevance of the average surface and cross-sectional properties of the films versus the crucial nano-structural effects of transiting to AKD coating, respectively. These two properties provide the balance between surface smoothness decreasing contact angle and surface nano-structure developed adopting the hydrophobising AKD acting to increase the contact angle, respectively.

The morphology of surface and cross-sections of the films with different mass ratios are illustrated by the SEM images in Figure 4, showing the distribution of the particles in the regenerated cellulose films and the cellulose matrix before and after the modification with AKD. It is readily apparent from Figure 4i–iv that pure cellulose film (V) has a smooth flat surface. When AKD is added, the surface of cellulose film becomes much rougher and hierarchical microstructures also appear (CA) (Figure 4v). The hierarchical structures could well account for the microcrystalline structural changes seen by XRD.

The wax-like particles of AKD, observed below the melt temperature of AKD, however, appear well-dispersed in the cellulose matrix. Furthermore, the roughness level increased along with the application of AKD coating, and this remains in reasonable agreement with previous literature [38,40]. Several pores can be observed on the surface of the coated CA-SO+PAO film (Figure 4vi). The surfaces of AKD-coated composite films are significantly rougher than uncoated film, especially the film containing 0.1% AKD (CA) (Figure 4v), and it is this specific roughness that can assist in reaching the ultimate target of superhydrophobicity.

However, after applying the coating (CA-SO+PAO+PAO) (Figure 4xi,xii), the voids of the regenerated cellulose films were covered by AKD, as presumed earlier, leading to a denser overall structure of the film [42]. The FESEM cross-section images of the films after coating clearly reveal higher density.

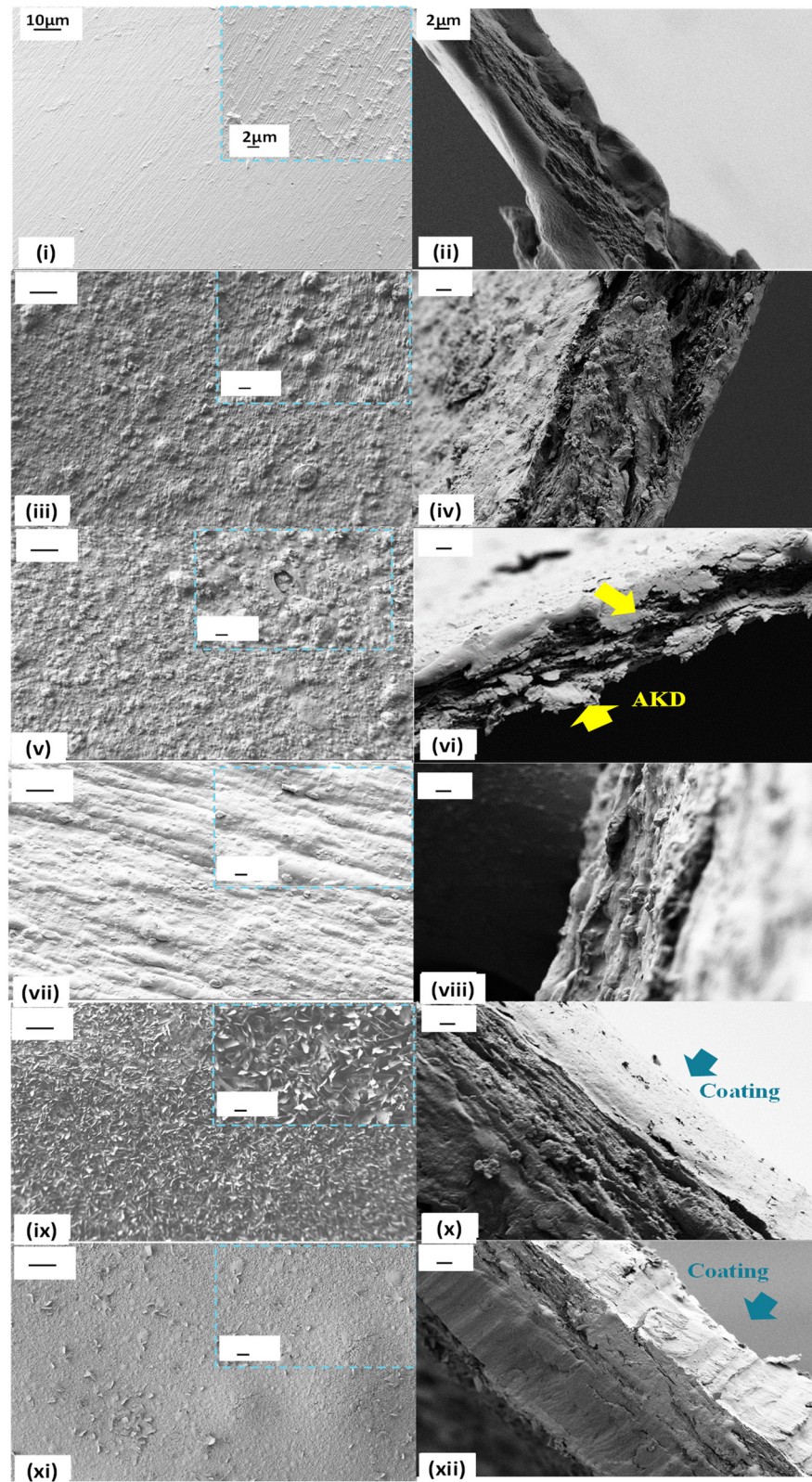

**Figure 4.** SEM images of the films, surface, cross-section films respectively, (**i**,**ii**) virgin cellulose (V), (**iii**,**iv**) ground Copy Paper (C), (**v**,**vi**) ground Copy Paper containing 1% internal AKD (CA), (**vii**,**viii**) Copy Paper containing 0.5% AKD and 0.5% starch (CA-S), (**ix**,**x**) CA-S film single coated in AKD emulsion (CA-SO+PAO), and (**xi**,**xii**) film with second AKD coating in emulsion (CA-SO+PAO+PAO). [Note that the yellow arrows indicate the penetration of the first layer AKD coating into the surface voids, and the blue arrows indicate the uniformity achieved by the double-coating procedure].

The optical microscope images in Figure 5 were specially filtered individually to enable visibility of the various component distributions over a larger image area than shown in the SEM images of Figure 4. It is therefore possible to visualise the layer thicknesses associated with cellulose alone in the control sample, Figure 5i, the clearly defined PCC filler particles in Figure 5ii, and the contrast in density observed as AK is added internally, Figure 5iii. Associated surface profile variation can be readily observed in Figure 5iv and contrasted with the subsequent AKD single and double coatings in Figure 5v,vi, respectively.

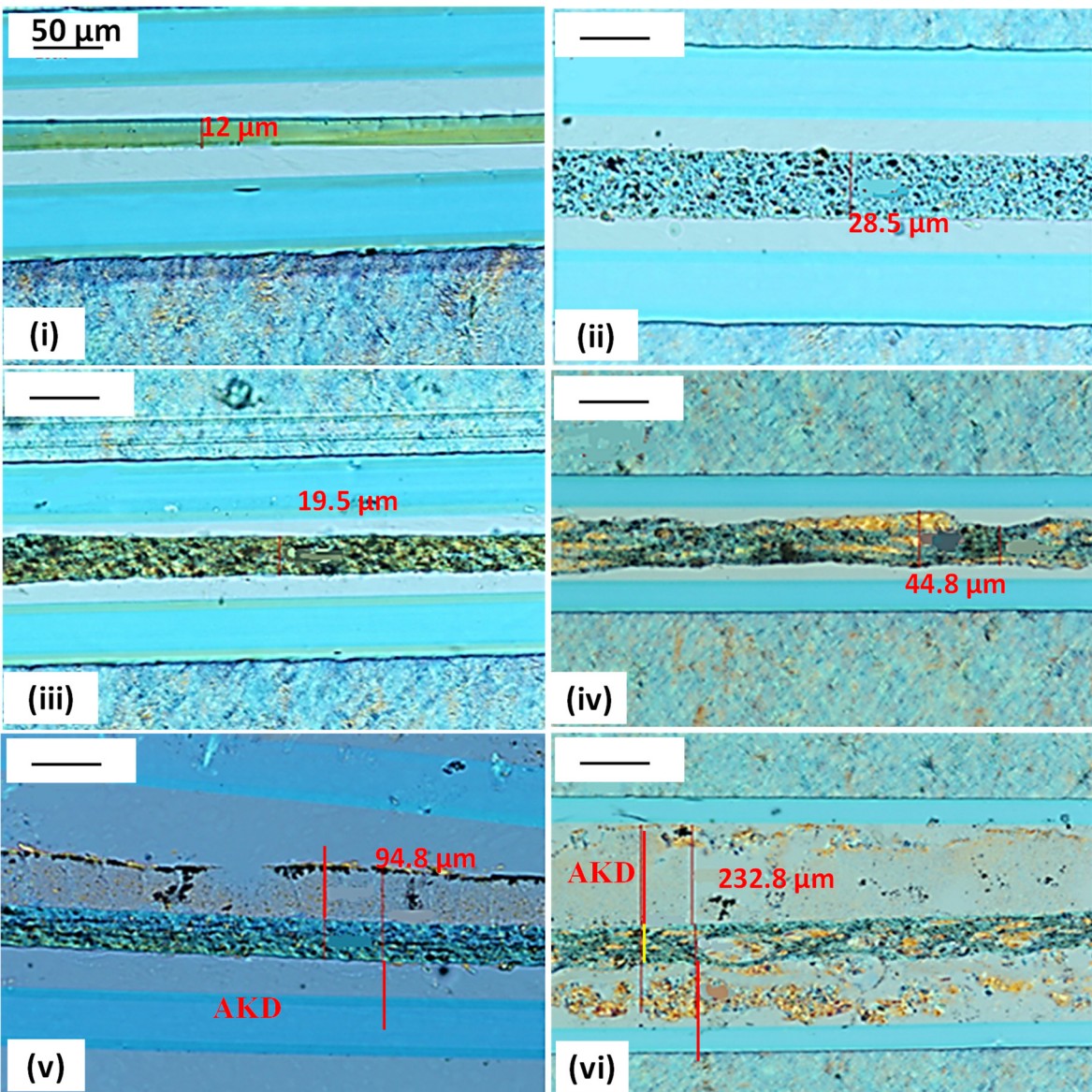

**Figure 5.** Optical microscopy of cross-section films respectively, (**i**) virgin cellulose (V), (**ii**) ground Copy Paper (C), (**iii**) ground Copy Paper containing 1% internal AKD (CA), (**iv**) Copy Paper containing 0.5% AKD and 0.5% starch (CA-S), (**v**) CA-S film single coated in AKD emulsion (CA-SO+PAO), and (**vi**) film with second AKD coating in emulsion (CA-SO+PAO+PAO). Layer regions of materials are shown by the vertical red lines, and their respective thickness is given. [Note that the scale bar in all images represents 50 μm. The red lines act to display the indicated material thickness].

## 3.2. Mechanical Properties of the Films

The mechanical properties of the regenerated pure ground virgin cellulose (V) and regenerated cellulose–calcium carbonate composite reinforced with a multilayer of AKD,

were measured by tensile testing at room temperature. Figure 6a shows the stress–strain curves of pure ground virgin cellulose and AKD cellulose composites films.

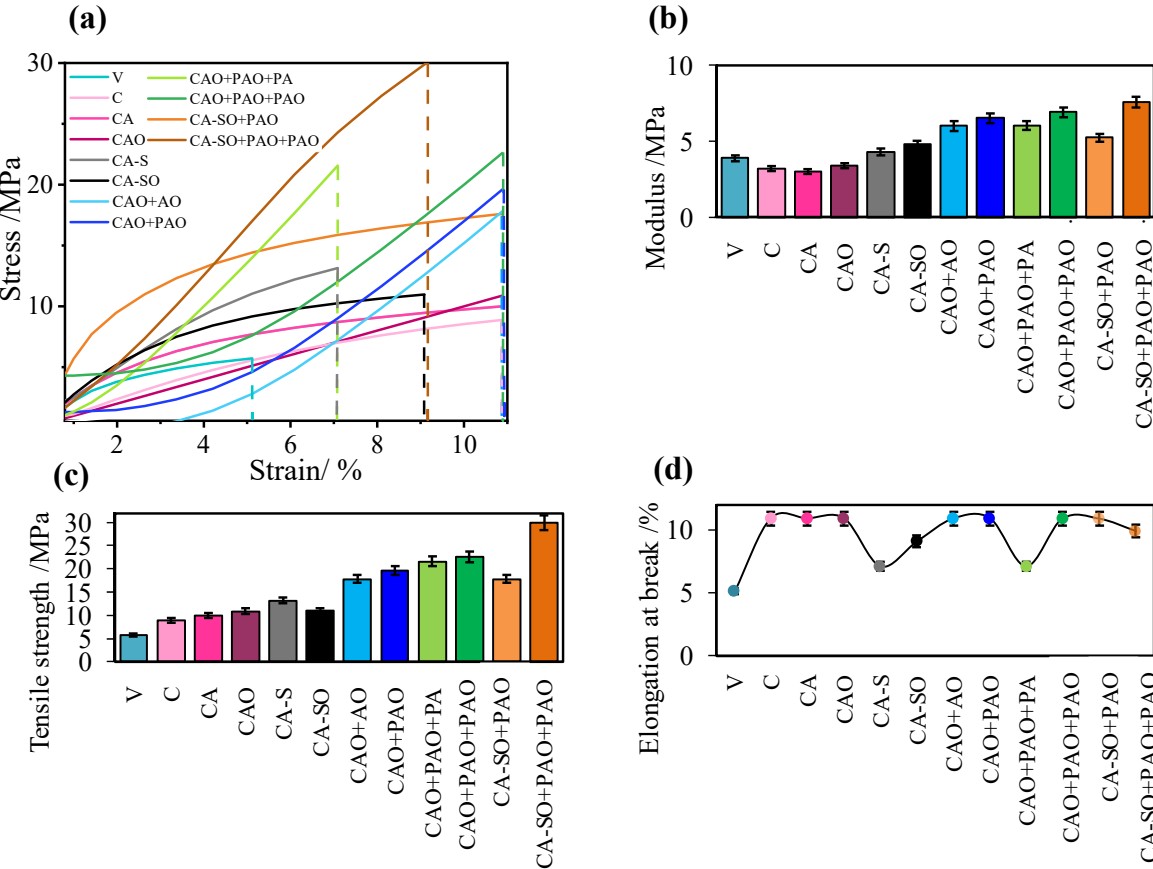

**Figure 6.** The mechanical properties of the films, including (**a**) representative tensile stress–strain curves, (**b**) Young's modulus, (**c**) tensile strength, and (**d**) elongation at break of the regenerated cellulose films treated with AKD.

Enhancement in the mechanical strength was observed when employing the starch into the film. This was ascribed to the adhesion by the starch in conjuncture with the cellulose. By adding the 0.5 *w/w*% starch into the cellulose-0.5 *w/w*% AKD dope when forming the film (CA-S), Young's modulus increased 42.66% compared with the similar film without starch having double the dose of AKD (1 *w/w*%) (CA) (Figure 6b). This finding aligns with Adibi et al. (2022); Li et al. (2021); Tian et al. (2022); Zou et al. (2021) [6,11,40,43] for their case of studying fibrillar cellulose films.

A schematic model of the novel coating procedure was shown in Figure 2. Adding the AKD layer as an external coating significantly increased further Young's modulus and tensile strength of the cellulose film (5.22 MPa), Figure 6b,c. This result indicates that a higher AKD content in the form of a laminate structure leads to better mechanical performance, suggesting that the AKD behaves classically as a reinforcement agent in the laminar system, i.e., acts as an engineering girder lattice reinforcement system. Double coating of AKD (CA-SO+PAO+PAO) gives an even greater increase in the Young's modulus (7.57 MPa).

Both the single and double coated samples show also high tensile strengths of 17.80 MPa and 29.89 MPa, respectively, as shown in Figure 6c. While polymer coatings in general are typically less stiff than the uncoated film substrate, they can act as adhesives on the film and enhance the strength per unit area. It is evident from Figure 6c that applying the AKD coating layer on the base cellulose film has resulted in slightly higher tensile strength than the uncoated film.

The lowest elongation at break was for the reference pure cellulose sample (V) at about 5.11% (Figure 6d). The effect of filler present typically increases this, as is the case here. AKD coating did not improve elongation at break for the coated films; however, heat treatment after coating increases it significantly. The mechanical results agree with Li et al. (2021); Tian et al. (2022) [40,43] but, as would be expected for composites, are in contrast to the results of Van Nguyen et al. (2022); Tarrés et al. (2018) [10,24,43].

### 3.3. Changing from Hydrophilicity to Reach the Level of Superhydrophobicity

Contact angle variation between the various sample formulations relates to the various hydrophobic surface energy components and surface smoothness and distribution of surface roughness present. To elucidate fully on particular values would require a more in-depth analysis of these parameters. We confine ourselves here to seeking the most effective combination to reach highest contact angle.

Since the surface hydrophobicity plays an important role in achieving barrier performance, the contact angle of the regenerated films for a water droplet on the surface should be raised as high as possible. The corresponding images of the water droplet were captured after 15 s. The static water contact angles on the different film surfaces are revealed in Figure 7a,c, where $\theta_{WCA}$ is the contact angle between the droplet of water and the film sample measured in air.

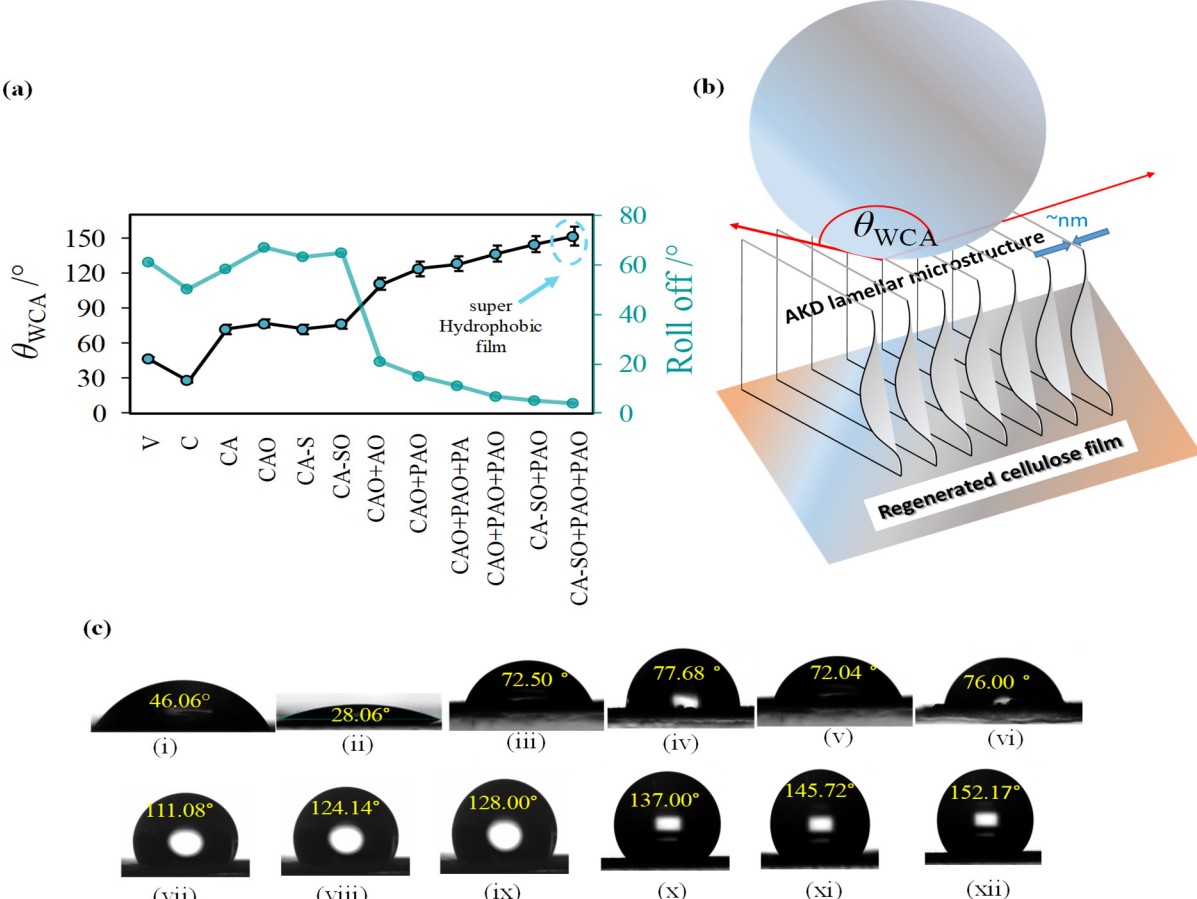

**Figure 7.** (**a**) Water contact angle ($\theta_{WCA}$), and Roll-off Angle ($\theta_{RoA}$), (**b**) schematic of a water droplet in Cassie-Baxter unstable equilibrium in contact with the novel AKD lamellar structure induced after solid precipitation onto precooled cellulose substrate from the heated aqueous emulsion and subsequent oven drying, and (**c**) contact angle of uncoated and coated films, (i)–(xii), V, C, CA, CAO, CA-S, CA-SO, CAO+AO, CAO+PAO, CAO+PAO+PA, CAO+PAO+PAO, CA-SO+PAO, and CA-SO+PAO+PAO, respectively.

It can be concluded that, first, the higher surface roughness can increase the hydrophobicity. For example, correlating with the SEM images (Figure 4), the water contact angle on the reference copy paper derived film, (C), with its smoother surface (Figure 4vii) was $72.50 \pm 2.01°$, already surprisingly high compared with the strong hydrophilicity of mechanically produced microfibrillated cellulose powder (for example, <20° [39] but which is, nonetheless, significantly lower than on the films showing greater roughness.

It is known that sizing is applied in the manufacture of copy paper, and this probably accounts for the reduced hydrophilicity of this, as yet here untreated, sample. Second, the more hydrophobic chemical(s) are applied, internally or as surface treatment, logically, the higher the contact angle obtained.

Following the observations above, the increasingly hydrophobic properties of the regenerated AKD containing composite films are manifested by the change in the water contact angle ($\theta_{WCA}$) being dependent on the AKD loading (Table 1). The pure ground virgin cellulose (V) and ground unprinted copy paper (C) (Figure 7a,b) were highly hydrophilic because of the abundance of hydroxyl groups in the cellulose structure, and relatively hydrophilic, respectively. Heat treatment changed the inherent hydrophilicity of cellulose films containing AKD.

For example, the cellulose film containing internal AKD (CA), the starch/cellulose film containing internal AKD and starch (CA-S), and coated cellulose/AKD with single layer AKD (CAO+PAO) each displayed slightly improved hydrophobicity after heat treatment, 6.94%, 5.55%, and 7.03%, respectively, compared to untreated films. In the case of CA-SO, the level of internal AKD is reduced due to the replacement by starch: this is not as efficient in producing hydrophobicity as the higher level of AKD alone.

As can be observed, the CAO-AO (Figure 7c(vii)), and CAO+PAO films (Figure 7c(viii)) are hydrophobic with $\theta_{WCA} = 111.08 \pm 4.33°$ and $124.14 \pm 2.90°$, respectively. In the case of CAO-AO there is a balance between the effect of generating a smoother surface due to applying a surface coating treatment, thus reducing contact angle, and the action of the treatment itself increasing hydrophobicity. The smoother surface effect in this case is greater than that of hydrophobising.

Pre-cooling of the film before hot AKD emulsion coating can also bring benefits of better coverage by the AKD, and this provides a marked further enhancement of hydrophobicity (11.71%). Notably, the increase in AKD loading from the basis regenerated CA film, having only internal AKD (uncoated and without oven drying), to that of the CAO+PAO+PAO film, with internal AKD, oven-dried and then double AKD coating applied with further oven drying, results in a change of basis weight from $33.98 \pm 5.68$ to $68.88 \pm 1.87$ g m$^{-2}$ (Table 2) and a parallel increase of $\theta_{WCA}$ from hydrophilic to superhydrophobic, i.e., from $\theta_{WCA} = 28.06 \pm 2.01°$ to $137.00 \pm 0.12°$.

Of most interest is the change between single and double AKD-coated films with oven drying, whilst containing starch and AKD in the cellulose dope, where the water contact angle significantly increases to $\theta_{WCA} = 145.72 \pm 0.12°$ (CA-SO+PAO) (Figure 7c(xi)) and $152.17 \pm 1.45°$ (having a roll-off tilt angle <5°) (CA-SO+PAO+PAO) (Figure 7c(xii)), respectively. This means that coating with AKD and oven drying changed the surface chemistry of the internally sized films from hydrophilic to superhydrophobic by the combined introduction of hydrophobic functional groups and surface microstructure, the former being aligned with the findings of Fedorov et al., 2020; Reverdy et al., 2018; Shen et al., 2019; Ryu et al., 2020; Yook et al., 2020 [22,23,37,42,44], and the latter being a novel finding reported in this work.

As can be seen in (CA-SO+PAO+PAO) Figure 7c(xii), the value of $\theta_{WCA} > 150°$ indicates superhydrophobicity. Combined with a roll-off tilt angle <5° for the water droplet this outstanding property could be confirmed.

The key factor differentiating between simply hydrophobising the surface and forming a superhydrophobic surface is, therefore, the nanoscale discontinuous structure formation in addition to the dispersive surface energy. These nanostructural discontinuities form upon oven drying after the AKD coating is applied. The growth of microlamellae of AKD

oriented perpendicular to the planar film surface results in an outer surface that consists of the essentially 2D nanometres-thin platelet edge exposed at the solid–air–liquid interface. This results in a suspension of the water droplet above the air-filled subsurface microstructure in a Cassie–Baxter minimum interface contact equilibrium as shown schematically in Figure 7b.

### 3.4. Barrier Properties of the Films

Having prevented liquid water transmission through the film by making the surface superhydrophobic, three of the most important remaining characteristics to achieve a functioning barrier layer in packaging are the prevention of water vapour transmission, to avoid or at least minimise moisture transfer exchange between the content, and the surrounding atmosphere; to stop gases, such as oxygen, passing into a vacuum or inert packed material or food; and a proofing against the passage of oils and fats for containing fatty foods and liquids.

#### 3.4.1. Water Vapour Transmission Rate

The water vapour transmission rate (WVTR) must be held as low as possible [1,45]. As shown in Figure 8a, the WVTR value of the copy paper film C (no internal AKD), and CA film (1% AKD internal) ranged from 26.66 to 8.24 g m$^{-2}$ d$^{-1}$. The vapour diffuses as a gas through the cellulose polymer membrane and hydrophilic OH groups can provide an additional driving force for water molecules to condense in the film matrix. Chemical modification of the cellulose films and hydrophobic coating reduce the water vapour transmission rate.

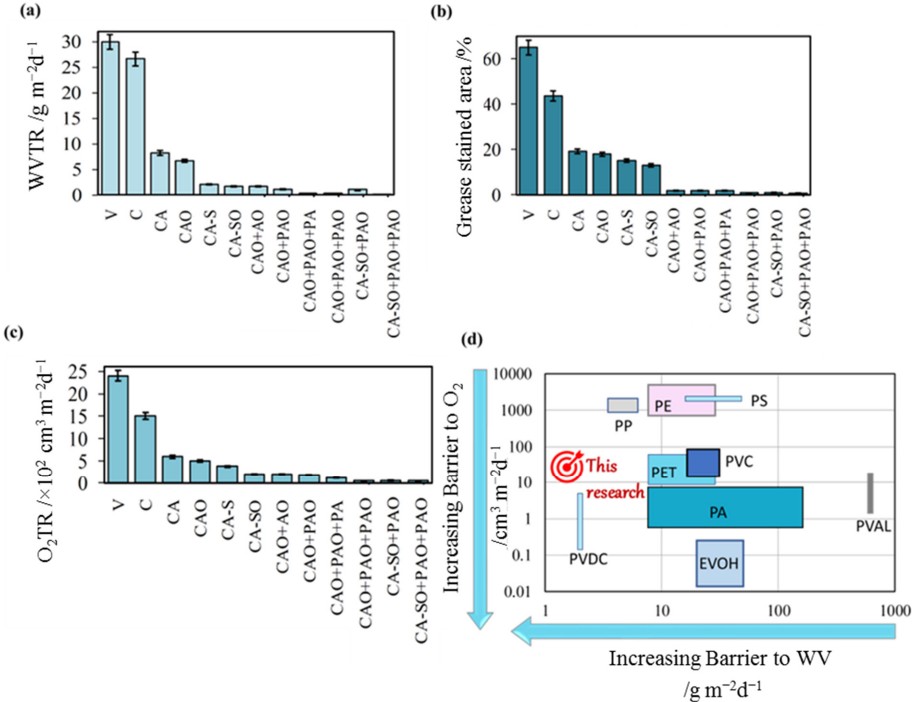

**Figure 8.** (**a**) The water vapour transmission rate (WVTR), (**b**) grease resistance of the films, (**c**) oxygen transmission rate (O$_2$TR), and (**d**) comparing the resistance against WVT and O$_2$T of the regenerated cellulose film treated with AKD in this study with those of some common polymers used in food packaging, ((**d**) is adapted with permission from [45]; Copyright year 2022; Copyright owners Nguyen et al.).

The hydrophilisation arising from regenerating the cellulose from IL clearly promotes water vapour transmission, and this must be countered using the hydrophobic film forming properties of starch within the dope and hydrophobising properties of AKD in the coating layer [44]. From Figure 8a, it is seen that the WVTR values in regenerated cellulose–calcium

carbonate composite-based films are decreased in comparison with the reference film containing virgin cellulose only (V). The precipitated calcium carbonate (PCC) added to the film component also decreased the water vapour transmission rate due to the increased tortuosity and, hence, the path length the water molecules must travel in their random walk diffusion around the mineral fillers.

The AKD hydrophobised the cellulose film, which was confirmed by their $\theta_{WCA}$ values (CA-SO+PAO) (Figure 7c(xi)). AKD also improves the water-vapour resistance of the film significantly, especially both the double dip-coated AKD-starch-containing oven-dried cellulose film (CA-SO+PAO+PAO) (WVTR 0.142 g m$^{-2}$ d$^{-1}$ being 92.90% lower than the uncoated reference (CA-S) (WVTR 2 g m$^{-2}$ d$^{-1}$), and the double dip coating cellulose AKD containing only the oven-dried film (CAO+PAO+PAO) (WVTR 0.260 g m$^{-2}$ d$^{-1}$) was further decreased to 96.84% lower than its relevant reference (CA) (WVTR 8.24 g m$^{-2}$ d$^{-1}$).

### 3.4.2. Grease/Oil Resistance

As can be seen in Figure 8b, the results show that the percentage ratio of the stained area to the unstained surface of contacted blotting paper for the samples V film (control sample), C film, and CA film are 65%, 43.66%, and 19.24%, respectively. As expected, these uncoated films showed poor to relatively poor grease transmission resistance. Interestingly, when starch was combined with cellulose instead of AKD alone, the grease resistance was slightly enhanced, displaying a change from a stained area of 19.24% (CA), for AKD-only-containing film, down to 15% (CA-S) for the AKD-starch-containing film. We suspect that this could be due, at least partially, to the natural vegetable oil already retained in maize starch, and thus there is no further capacity for absorption, whereas AKD is widely reported to exhibit selective oil removal due to its promotion of high oil absorptivity [46] (Delgado-Aguilar et al., 2016). In addition, as observed in Figures 4 and 5 and Table 2, uncoated films are significantly more porous, a fact that promotes the entrance of grease into the film structure [10].

In contrast, double AKD coating on AKD-starch/cellulose film (CA-SO+PAO+PAO), and AKD-cellulose film (CAO+PAO+PAO) effectively present a complete proofing against oil/grease, which is aligned with the findings for cellulose nanofibril (CNF) films [42]. This result reveals the synergistic effect between the additive components in the base film and the coating formulation.

Mechanistically, the grease/oil barrier property associated with the use of otherwise hydrophobising and, thus, tendentially lipophilic components might seem contradictory. The hydrophobisation of regenerated cellulose film suggests that some cellulose hydroxyl groups are substituted by aliphatic hydrophobic chains. Since the intrinsic hydrophilicity and hydrogen bonds have been suggested to provide the grease barrier properties of cellulose films, the presence of long alkyl groups may cause a loosening effect in the hydrogen bonds between cellulose in the base film. This effect could act in addition to the somewhat high permeability of the hand-made films being studied here to lead to a certain oil absorption effect in the case of uncoated internally hydrophobised films. Therefore, the result that application of the denser AKD coating results in a significantly greater grease resistance compared with that of the uncoated films, likely relates to the action of the AKD coating layer(s) covering the surface entry pores [6].

Subsequent heat treatment can lead to liquefaction of the AKD and thus an additional internal blocking of some of the pores deeper in the bulk film structure, thereby, reducing the pore connectivity and inducing a more tortuous path for the oil molecules, similar to the findings in the case Yook et al., 2020 [44]. This interpretation can be supported by comparing the microscopic images of the surface and cross-section of the uncoated and coated samples in Figures 4 and 5.

Based on SEM images (Figures 4 and 5) and Table 2, coating thicknesses of 93.80 ± 1.10 and 129.33 ± 9.40 μm, respectively, each for one-layer AKD emulsion coatings on cellulose film (CAO+AO) and starch/cellulose film (CA-SO+PAO) can be compared to the double layers on cellulose film (CAO+PAO+PAO) and AKD-starch/cellulose film (CA-

SO+PAO+PAO), having coating thicknesses of 216.01 $\pm$ 7.23 and 211.21 $\pm$ 6.25 µm, respectively. The resulting greatly reduced void size and porosity effectively reflect the blockage of the micro/nano-flow paths for both grease and water molecules [27].

### 3.4.3. Oxygen Gas Permeation Properties

Oxygen-permeation barrier properties (OP) of the regenerated cellulose films were determined at ambient 23 °C temperature and 50% RH. By blocking oxygen from entering the package, one can improve the shelf life of food by delaying its degradation. $O_2$ permeability, quoted as the $O_2$ transmission rate ($O_2$TR), decreased by about 60% on the addition of AKD (1% AKD) internally, as can be seen, comparing the uncoated cellulose film C with the uncoated CA film in Figure 8c.

It is clear also in Figure 8c that the single AKD coating (CAO+PAO) significantly improved the oxygen barrier capabilities above that of the substrate film alone. The $O_2$ permeability decreased further with the increase in coat weight and further modification of the regenerated cellulose films, e.g., a noticeable enhancement in the oxygen barrier is observed between CAO+PAO ($1.85 \times 10^2$ cm$^3$ m$^{-2}$ d$^{-1}$) and CAO+PAO+PAO ($0.5 \times 10^2$ cm$^3$ m$^{-2}$ d$^{-1}$), i.e., enhancements of about 69.16% and 91.66%, respectively, compared with the CA film alone (uncoated).

The greatest reduction in the oxygen transmission rate was observed with a double layer coating of AKD applied on the CA-SO+PAO+PAO ($0.45 \times 10^2$ cm$^3$ m$^{-2}$ d$^{-1}$) compared to uncoated film ($3.85 \times 10^2$ cm$^3$ m$^{-2}$ d$^{-1}$). Thus, using an aqueous, heated AKD emulsion applied onto a precooled film is an enhanced way to achieve effective oxygen barriers. As described above for grease barrier effects, by coating an AKD emulsion on top of the precooled film, the molten polymer acts to fill any pinholes/defects present in the film [6], thereby, resulting in a steep decrease and lower scatter in $O_2$TR values (Figure 4xii). Furthermore, when an AKD emulsion is coated on an AKD-cellulose film, hydroxyl groups in the cellulose film that might have remained unreacted with the internal AKD may be available to form networks that interfere with oxygen transmission [44].

Importantly, the oxygen transmission rate through all the filler-containing films was observed to be less in comparison to the virgin cellulose film (V) at constant pressures [41]. The dispersion of precipitated calcium carbonate (PCC) within the contrasting composite film provides a hurdle in the form of a tortuous path to oxygen entrance and passage through the film, whereas virgin cellulose may have regularly ordered connected voids providing for greater oxygen permeation.

In summary, the grease and water vapour barrier properties that are crucial to a successful barrier development can be achieved by combining the film-forming behaviour of cellulose derived from the process of forming an IL solution dope, enhanced by the inclusion of calcium carbonate filler, generating micro- and nanosized composite crystallite overlap on the regenerated cellulose film surface, similar to that seen for the fibrils in CNF film (He et al., 2021) [7], by including hydrophobising additives.

Further enhancement can be achieved using the novel coating technique for AKD reported in this work. The resulting bio-based AKD-coated cellulose film regenerated potentially from mineral filled paper waste meets the barrier properties of petroleum-based substrates in respect to the $O_2$TR and WVTR requirements for most groups of food packaging, and is performance competitive with conventional polymeric packaging (Figure 8d).

We recognise there is a difficulty in assessing the overall degree of success that the novel composite material reaches in respect to the barrier film properties of equivalent composites. This is because there are no references available to compare IL dissolved cellulose being developed as a barrier film in this way by adding hydrophobising and permeation-blocking additives to the IL dissolution process.

What we provide to assist in critical assessment is to refer the reader to the review data in Figure 8d, in which existing industrially accepted barrier materials made from oil-based polymers are compared with our results. We show that we reached values equivalent to PET and PVC for $O_2$ barrier and superior values to all reported barrier materials in the

same review in respect to preventing water vapour transmission. We hope that, by drawing attention to the novelty and the comparison with referenced barriers in use today, we can convincingly display the merit of the material achievement.

### 3.5. Reaching the Practical Barrier Property Target-Product Surface Robustness

One final criterion is the physical surface durability/robustness. As reported by Kostic et al., 2022 [16], the mechanical properties of the regenerated substrate films can also match those of polyethylene and polypropylene. The surface robustness of the AKD coatings used here, nonetheless, could pose a challenge.

It can be concluded from the results of the Taber abrasion test, shown in Figure 9a, that the abrasion resistance of the uncoated samples is, as expected, higher than for the coated ones. The abrasion resistance decreased further with the increase in coat weight and further modification of the regenerated cellulose films, e.g., especially the double dip-coated AKD-starch-containing cellulose film (CA-SO+PAO+PAO) (76.15%) was 23.15% lower than the reference uncoated film (CA-S) (99.1%), and the double dip-coated cellulose AKD-only-containing film (CAO+PAO+PAO) (75%) was 27.85% lower than its reference (CA) (99.27%).

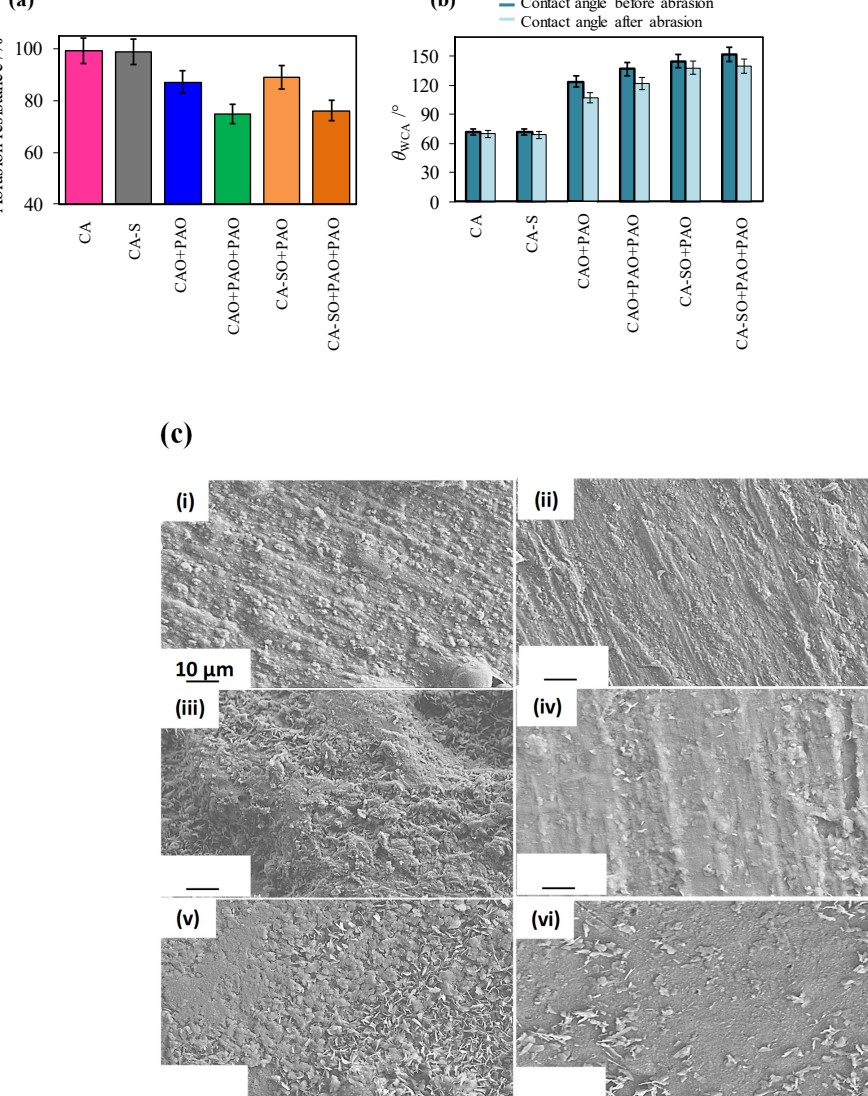

**Figure 9.** (**a**) Abrasion resistance of the comparative reference and coated films, (**b**) static contact angles after abrasion, and (**c**) SEM images of the surface layer after 20 cycles of Taber abrasion, CA (i), CA-S (ii), CA+PAO (iii), CA+PAO+PAO (iv), CA-SO+PAO (v), and CA-SO+PAO+PAO (vi).

However, these abrasion values still indicate significant retention of the layer under the relatively severe conditions in the test. This is confirmed both by the contact angle observations before and after abrasion (Figure 9b) and by the SEM images in Figure 9c, where the fundamental surface fine-scale structure can be seen to have been retained with evidence of abrasion shown as wear marks superposed onto the underlying fine structure. The contact angle naturally drops, albeit only slightly, after surface wear has occurred as the surface peaks of the lamellar structure become partly blunted and abraded; however, the superhydrophobicity is still remarkably maintained within the measurement error.

Whilst presenting some limited information on the effects of abrasion, we recognise that the result is only illustrative—considered here under a very small range of abrasion and environmental conditions and applied only to hand-made laboratory samples. Therefore, seeking to comment on the whole range of property changes due to abrasion beyond that of the contact angle, such as the water vapour transmission and grease resistance, we consider to be a matter for future development work.

It can be expected that improvements in the film uniformity beyond the hand-made laboratory scale will go a long way towards increasing product robustness as well as reducing the permeability to liquids, vapours, and gases, and in turn reducing the need for relatively thick coatings of hydrophobising chemicals. Future study of the potential for adding strengthening mineral filler to the coating layer, similar to the calcium carbonate in the cellulose film itself, could well be profitable in terms of the product surface robustness as well as cost reductions.

## 4. Conclusions

One of the major challenges for cellulose films, including those made in recent times from cellulose nanofibrils (CNF), is their intrinsically poor resistance to wetting by water and high water vapour transmission. A recently published novel approach was applied here regarding regenerating a cellulose–calcium carbonate composite film using an ionic liquid dissolution of waste paper (copy paper), which demonstrated mechanical strength equivalent to oil-based polymer films [16].

Given the many demands on packaging performance, including strength versus plasticity versus elasticity, smoothness versus roughness, hydrophobicity versus hydrophilicity, and barrier versus breathable film, we intended to illustrate the broad range that the concept described in this work provides. However, to emphasise a single goal, the ability to reach superhydrophicity together with barrier properties can be singled out as a major technological step.

We demonstrated here that the resulting mechanically strong biomass-based, biodegradable laminate film composite structure exhibited excellent barrier performance against water vapour and oxygen as well as a being a liquid barrier against water and grease/oil and thus could serve to replace oil-based polymer primarily in the packaging industry. These could also be developed to grades suitable for construction and agriculture—including plant active nutrients. Such a step additionally provides a complete circular economy solution to waste biomass and mineral fillers, thereby, enhancing sustainability and ultimately eliminating non-biodegradable plastic waste.

**Author Contributions:** Conceptualization, P.G. and M.I.; methodology, P.G., M.I., N.B. and J.L.; validation, P.G., M.K. and A.I.; formal analysis, P.G., M.I. and M.K.; investigation, M.I., K.D.-M., M.K., A.I. and N.B.; resources, P.G., P.U., D.J., M.K. and E.B.; data curation, M.I.; writing—original draft preparation, M.I. and P.G.; writing—review and editing, M.I., P.G., M.K. and A.I.; visualization, M.I. and P.G.; supervision, P.G.; project administration, P.G., P.U., D.J., M.K. and E.B.; funding acquisition, P.G. and E.B. All authors have read and agreed to the published version of the manuscript.

**Funding:** The work was supported by Omya International AG, Switzerland, Group Sustainability.

**Institutional Review Board Statement:** Not applicable.

**Informed Consent Statement:** Not applicable.

**Data Availability Statement:** Not applicable.

**Conflicts of Interest:** The authors declare no conflict of interest.

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
