# Peer review of "Achieving a Superhydrophobic, Moisture, Oil and Gas Barrier Film Using a Regenerated Cellulose–Calcium Carbonate Composite Derived from Paper Components or Waste"

_sustainability, doi:10.3390/su141610425_

Round 1

Reviewer 1 Report

The manuscript entitled "Achieving a superhydrophobic, moisture, oil and gas barrier film using regenerated cellulose-calcium carbonate composite derived from paper components or waste" by Imani et al. basically deals with the preparation of cellulose-calcium carbonate composite films from waste/ residues which have a circular economy and ecofriendly approach. However, the manuscript lacks coherence in many places as follows which must be amended for further consideration:

1. The "Abstract" section is excessively exaggerated with unnecessary detailing! Keep it composed and brief.

2. The "Introduction" section lacks coherence in different paragraphs! It must be a story overall, which is lacking. The Introduction section also totally lacks the hypotheses and aim of the study! This must be added in the last part of the introduction like any other standard publication. Line no. 88 has unnecessarily many references without much detailing, please take care of that.

3. The methodology totally lacks citations, which must be improved.

4.  Please provide unedited resolution of optical microscopy images of Fig. 5! Too much colour contrast changes have been made.

5. The manuscript totally lacks discussion with the relevant publications. Please do compare your result with others in the same field. Also, why the authors are claiming the work is novel is neither clear! Why a separate section "summary" is added is not understandable too. Please revise the manuscript like any other standard scientific manuscript format.

6. The "Conclusion" section is not like a standard "conclusion" section of standard scientific manuscripts. Please conclude the important finding of your result in this section and give only a brief of work to be done (if needed) and future prospects. The summary can be a part of the conclusion instead.

Author Response

Response to Reviewers' Comments:

Ref: sustainability-1809910, "Achieving a superhydrophobic, moisture, oil and gas barrier film using regenerated cellulose-calcium carbonate composite derived from paper components or waste"

We acknowledge the suggestions provided by the reviewers as well as those from the editorial office. All the items are addressed in the revised version of the manuscript. The itemised list below incorporates all comments and point-by-point reply to each item. The corresponding changes are highlighted in the revised manuscript using the “Track Changes” function provided for further processing. We appreciate the suggestions provided by the reviewers and editor, who volunteered insightful comments that definitely helped to enhance the quality and impact of our contribution.

Reviewer 1

Open Review #1

English language and style

( ) Extensive editing of English language and style required
( ) Moderate English changes required
(x) English language and style are fine/minor spell check required ( ) I don't feel qualified to judge about the English language and style

Yes

Can be improved

Must be improved

Not applicable

Is the content succinctly described and contextualized with respect to previous and present theoretical background and empirical research (if applicable) on the topic?

( )

(x)

( )

( )

Are all the cited references relevant to the research?

(x)

( )

( )

( )

Are the research design, questions, hypotheses and methods clearly stated?

(x)

( )

( )

( )

Are the arguments and discussion of findings coherent, balanced and compelling?

( )

(x)

( )

( )

For empirical research, are the results clearly presented?

(x)

( )

( )

( )

Is the article adequately referenced?

(x)

( )

( )

( )

Are the conclusions thoroughly supported by the results presented in the article or referenced in secondary literature?

( )

(x)

( )

( )

Comments and Suggestions for Authors

In this manuscript, the authors use eco-friendly, non-toxic and cost-effective alkyl ketene dimer (AKD) and starch to achieve barrier properties of novel cellulose-calcium carbonate composite films regenerated from paper components using ionic liquid as solvent. In general, this is an interesting work, below are some questions and comments that need to be addressed.

Thank you for the positive commentary on the work.

  1. In the caption of Fig. 3, please add more detailed explanation of the FTIR spectra and XRD data.

***ANSWER: Done as suggested with further information provided. 

  1. In Fig. 7(a), there is a significant drop in contact angle for CA-SO and CAO-AO, what is the underlying rationale for this?

***ANSWER: In the case of CA-SO, the level of internal AKD has been reduced due to the replacement by starch: this is not as efficient in producing hydrophobicity as the higher level of AKD alone. In the case of CAO-AO there is a balance between the effect of generating a smoother surface due to applying a surface coating treatment, thus reducing contact angle, and the action of the treatment itself increasing hydrophobicity. The smoother surface effect in this case has been greater than that of hydrophobising.

  1. What is the lowest thickness of the film that could maintain the good reported mechanical property?

***ANSWER: The thinnest film properties are dependent largely on the quality of production. Here we stress that the films are hand-made and no claim of perfection in quality is made. The thinnest film in the series studied was 15.75±5.72 µm and we would not predict values for films thinner than this using the laboratory film forming method applied. However, ultimately, the largest filler particle used would define the thinnest film that could be made uniformly.

  1. After the abrasive test, would the film still maintain the good permeability and mechanical properties as reported?

***ANSWER: We appreciate your valuable comment. In response to your recommendation, we have added Figures 9(b), and (c), respectively, in our manuscript also referred in the text.

Reviewer 2 Report

In this manuscript, the authors use eco-friendly, non-toxic and cost-effective alkyl ketene dimer (AKD) and starch to achieve barrier properties of novel cellulose-calcium carbonate composite films regenerated from paper components using ionic liquid as solvent. In general, this is an interesting work, below are some questions and comments that need to be addressed.

1. In the caption of Fig. 3, please add more detailed explanation of the FTIR spectra and XRD data.

2. In Fig. 7a, there is a significant drop in contact angle for CA-SO and CAO-AO, what is the underlying rationale for this?

3. What is the lowest thickness of the film that could maintain the good reported mechanical property?

4. After the abrasive test, would the film still maintain the good permeability and mechanical properties as reported?

Author Response

Response to Reviewers' Comments:

Ref: sustainability-1809910, "Achieving a superhydrophobic, moisture, oil and gas barrier film using regenerated cellulose-calcium carbonate composite derived from paper components or waste"

We acknowledge the suggestions provided by the reviewers as well as those from the editorial office. All the items are addressed in the revised version of the manuscript. The itemised list below incorporates all comments and point-by-point reply to each item. The corresponding changes are highlighted in the revised manuscript using the “Track Changes” function provided for further processing. We appreciate the suggestions provided by the reviewers and editor, who volunteered insightful comments that definitely helped to enhance the quality and impact of our contribution.

Reviewer 2

Open Review #2

English language and style

(x) Extensive editing of English language and style required
( ) Moderate English changes required
( ) English language and style are fine/minor spell check required
( ) I don't feel qualified to judge about the English language and style

Yes

Can be improved

Must be improved

Not applicable

Is the content succinctly described and contextualized with respect to previous and present theoretical background and empirical research (if applicable) on the topic?

( )

( )

( )

(x)

Are all the cited references relevant to the research?

( )

(x)

( )

( )

Are the research design, questions, hypotheses and methods clearly stated?

(x)

( )

( )

( )

Are the arguments and discussion of findings coherent, balanced and compelling?

( )

(x)

( )

( )

For empirical research, are the results clearly presented?

(x)

( )

( )

( )

Is the article adequately referenced?

( )

( )

(x)

( )

Are the conclusions thoroughly supported by the results presented in the article or referenced in secondary literature?

( )

(x)

( )

( )

Comments and Suggestions for Authors

  1. There are too many keywords in this article and typically, the number of keywords should be limited to 5 to 6. Please choose the most important words as keywords.

***ANSWER: Modified as requested.

  1. The English level should be improved in order to express the meaning much more clearly. This is especially true for the abstract part and conclusion part. Authors are advised to ask an English-native speaker or a language editing company to help to modify the English in the text. For example, in the abstract part, in Line 35, “The initially high water vapour and oxygen transmission rates” should be modified as “The initial high water vapour and oxygen transmission rates”. Compound sentences, of course, can be used, while the meaning should be stated clearly. In line 27,28, however, the meaning is not that clearly stated, which should be corrected. Similar things should be checked throughout the whole text. There are many similar mistakes in the main text, especially in the introduction part. The summary part should also be modified carefully in order to clearly express its meaning.

***ANSWER: In response to the reviewer’s comment, we confirm that the paper has been thoroughly revised by a native English-speaking scientist.

  1. In the introduction part, the author mentioned that “A most relevant example has been the formation of a novel nanocomposite, together with mineral calcium carbonate, with strength properties similar to polyethylene and polypropylene, achieved by controlled recrystallisation on cooling developing a cellulose crystallite size ranging from 1.4 – 3.9 nm depending on calcium carbonate filler content and type [14,15]. ” Indeed, Minerals are good choice for the packaging materials. And among the minerals, calcium carbonate is not the best choice. Layered silicates, especially sericite, is the most promising one, due to its excellent chemical resistance, improved light stability, high tensile strength, high heat distortion temperature and excellent barrier properties. Sericite is abundant in nature and the price of it is low, making it very promising in the future. The relevant references are “(1) Liang, Y.; Ding, H.; Wang, Y.; Liang, N.; Wang, G. Intercalation of cetyl trimethylammonium ion into sericite in the solvent of dimethyl sulfoxide. Appl. Clay Sci. 2013, 74, 109–114; (2) Liang, Y.; Yang, D.; Yang, T.; Liang, N.; Ding, H. The Stability of Intercalated Sericite by Cetyl Trimethylammonium Ion under Different Conditions and the Preparation of Sericite/Polymer Nanocomposites. Polymers 2019, 11, 900”, which should be cited in the introduction part to make it more comprehensive.

***ANSWER: Yes, we fully agree that other mineral choices might give enhanced properties above those of using calcium carbonate in certain respects, due to particle morphology (aspect ratio) and surface energy criteria. However, we draw attention to the reviewer that we are specifically considering the opportunities of circular economy offered by using potential waste papers. The mineral example given by the reviewer is not prevalent in recycled or waste cellulose containing materials. By a huge margin, the mineral most prevalent in these waste products is calcium carbonate, both as filler and as coating pigment. Therefore, we illustrate this as the circular economy example for our novel concept. This is naturally the suitable target for developing sustainability. Nonetheless, we now state that if using virgin materials then alternative mineral might add extra performance in certain ways, though not in cost-brightness performance, as follows:

“Whilst choosing calcium carbonate as the exemplified filler, we acknowledge that certain barrier properties in one-dimensional transmission mode, such as permeation through films, could be enhanced more effectively using materials of platy morphology and high aspect ratio. Previous work by Liang et al. (2013, 2019) [35,36] has illustrated the advantageous properties of sericite, for example, whilst many have studied the use of high aspect ratio kaolin, montmorillonite, talc etc. [37] showing specific advantages of these materials. However, we stress the importance of calcium carbonate in this context directly related to its vastly greater prevalence in the source material envisaged in our circular economy proposal, namely, recycled and waste paper and board, including the dominance as waste mineral in deinking sludge arising from the paper and board recycling process. Our aim is to retain these otherwise materials inside the circular economy and prevent their disuse into land-fill etc. Finally, resistance to permeation is often sought-after in three-dimensions, such as in bulk volume construction block-like materials. In this case, it is the probability of collision between the permeating molecules and barrier particles that is critical, and this depends on particle number, related primarily to the ratio of particle fineness per unit weight addition. In this three-dimensional context, nano calcium carbonate is as effective as other readily available nano materials, as exemplified by the design of caps for PET bottles and in containers [38,39].

  1. The type setting of the whole article should be changed as a whole. For instance, subheadings of the article should be in similar places and two blank spaces should be set at the beginning of each paragraph. This should be paid attention specially.

***ANSWER: Many thanks for your comment. Modifications made as requested.

  1. As to the figures in Fig. 3, both the right lines of (a) and (b) are missing, which should be modified.

***ANSWER:  Thank you very much for your precise observation. We have modified the Fig. 3 accordingly.

  1. Why an underline is used in Line 508? If it is not useful, please delete it.

***ANSWER: There is no underlining in the original manuscript. Editorial highlighting, however, has been made by the journal to draw attention to matters of formatting etc. These will be dealt with during the proofing process.

  1. The “Summary” and “conclusion” part should be combined into one part.

***ANSWER: We have combined the two sections under Conclusions so as to follow the reviewer’s recommendation. As background, we specifically separated them in the original since many scientific authors unfortunately fail to understand that Conclusions actually should consist of the new implications and future effect of the work, not a simple summary.

  1. What are the differences between the SEM images and optical microscopy images of the prepared series samples? Can only one kind of data be kept?

***ANSWER: Good point raised – thank you.

The optical images contain individual material separation more clearly and provide a statistically more relevant impression of the average surface and cross-sectional properties, and so we request that they are retained. The SEM images are essential in contrast to illustrate the nano-structural effect of transiting to AKD coating, and so we request that these also are retained. We add the following text in support of this:

“The comparison made between optical and SEM images enables the viewer to establish a statistical relevance of the average surface and cross-sectional properties of the films versus the crucial nano-structural effects of transiting to AKD coating, respectively. These two properties provide the balance between surface smoothness decreasing contact angle and surface nano-structure developed adopting the hydrophobising AKD acting to increase the contact angle, respectively.”

  1. The abbreviation of the samples in data should be clearly stated so that readers do not need to search throughout the whole text to find its meaning.

***ANSWER:  We agree. However, reviewers often say the opposite that abbreviated labels should be established at the beginning and then only those used throughout. We are pleased in this case to retain more sample explanation.

  1. As to the abrasion part, can SEM images or similar images be put so that to show the morphology of the samples after abrasion to make a comparison?

***ANSWER: We appreciate the value of extending the observations for this topic. In response to your recommendation, we have added Figures 9(b), and (c), respectively, in our manuscript also referred to in the text. 

Submission Date

24 June 2022

Date of this review

11 Jul 2022 10:06:20

Reviewer 3 Report

1.    There are too many keywords in this article and typically, the number of keywords should be limited to 5 to 6. Please choose the most important words as keywords.

2.    The English level should be improved in order to express the meaning much more clearly. This is especially true for the abstract part and conclusion part. Authors are advised to ask an English-native speaker or a language editing company to help to modify the English in the text. For example, in the abstract part, in Line 35, “The initially high water vapour and oxygen transmission rates” should be modified as “The initial high water vapour and oxygen transmission rates”. Compound sentences, of course, can be used, while the meaning should be stated clearly. In line 27,28, however, the meaning is not that clearly stated, which should be corrected. Similar things should be checked throughout the whole text. There are many similar mistakes in the main text, especially in the introduction part. The summary part should also be modified carefully in order to clearly express its meaning.

3.    In the introduction part, the author mentioned that “A most relevant example has been the formation of a novel nanocomposite, together with mineral calcium carbonate, with strength properties similar to polyethylene and polypropylene, achieved by controlled recrystallisation on cooling developing a cellulose crystallite size ranging from 1.4 – 3.9 nm depending on calcium carbonate filler content and type [14,15]. ” Indeed, Minerals are good choice for the packaging materials. And among the minerals, calcium carbonate is not the best choice. Layered silicates, especially sericite, is the most promising one, due to its excellent chemical resistance, improved light stability, high tensile strength, high heat distortion temperature and excellent barrier properties. Sericite is abundant in nature and the price of it is low, making it very promising in the future. The relevant references are “(1) Liang, Y.; Ding, H.; Wang, Y.; Liang, N.; Wang, G. Intercalation of cetyl trimethylammonium ion into sericite in the solvent of dimethyl sulfoxide. Appl. Clay Sci. 2013, 74, 109–114; (2) Liang, Y.; Yang, D.; Yang, T.; Liang, N.; Ding, H. The Stability of Intercalated Sericite by Cetyl Trimethylammonium Ion under Different Conditions and the Preparation of Sericite/Polymer Nanocomposites. Polymers 2019, 11, 900”, which should be cited in the introduction part to make it more comprehensive.

4.    The type setting of the whole article should be changed as a whole. For instance, subheadings of the article should be in similar places and two blank spaces should be set at the beginning of each paragraph. This should be paid attention specially.

5.    As to the figures, in Fig. 3, both the right lines of (a) and (b) are missing, which should be modified.

6.    Why an underline is used in Line 508? If it is not useful, please delete it.

7.    The “Summary” and “conclusion” part should be combined into one part.

8.    What are the differences between the SEM images and optical microscopy images of the prepared series samples? Can only one kind of data be kept?

9.    The abbreviation of the samples in data should be clearly stated so that readers do not need to search throughout the whole text to find its meaning.

10. As to the abrasion part, can SEM images or similar images be put so that to show the morphology of the samples after abrasion to make a comparison?

Author Response

Response to Reviewers' Comments:

Ref: sustainability-1809910, "Achieving a superhydrophobic, moisture, oil and gas barrier film using regenerated cellulose-calcium carbonate composite derived from paper components or waste"

We acknowledge the suggestions provided by the reviewers as well as those from the editorial office. All the items are addressed in the revised version of the manuscript. The itemised list below incorporates all comments and point-by-point reply to each item. The corresponding changes are highlighted in the revised manuscript using the “Track Changes” function provided for further processing. We appreciate the suggestions provided by the reviewers and editor, who volunteered insightful comments that definitely helped to enhance the quality and impact of our contribution.

Reviewer 3

Open Review #3

English language and style

(x) Extensive editing of English language and style required
( ) Moderate English changes required
( ) English language and style are fine/minor spell check required
( ) I don't feel qualified to judge about the English language and style

Yes

Can be improved

Must be improved

Not applicable

Is the content succinctly described and contextualized with respect to previous and present theoretical background and empirical research (if applicable) on the topic?

( )

(x)

( )

( )

Are all the cited references relevant to the research?

(x)

( )

( )

( )

Are the research design, questions, hypotheses and methods clearly stated?

(x)

( )

( )

( )

Are the arguments and discussion of findings coherent, balanced and compelling?

(x)

( )

( )

( )

For empirical research, are the results clearly presented?

(x)

( )

( )

( )

Is the article adequately referenced?

(x)

( )

( )

( )

Are the conclusions thoroughly supported by the results presented in the article or referenced in secondary literature?

(x)

( )

( )

( )

Comments and Suggestions for Authors

The author described a kind of regenerated cellulosic composite film with multiple barrier properties such as moisture, water vapor, oxygen and oil resistance, for food packaging applications. The multi-functional cellulose film was prepared by modifying in bulk and surface. AKD is commonly used for hydrophobic modification of cellulose substrate. Although the novelty of this paper is mediocre, the work is relatively comprehensive, and the influence of different preparation processes on the properties of the films was discussed.

Thank you for the commentary on the work.

***ANSWER: We accept, and widely reference the fact, that AKD has been used for a long time to hydrophobise cellulose, but it has never been used in the context of incorporation during the unique dissolution via ionic liquid, nor has it been used in the case of such regenerated cellulose incorporating filler particles.

To emphasise this novelty we insert the following:

“Whilst we recognise, and widely reference that AKD has been used regularly to hydrophobise cellulose, it has, nonetheless, never been used in the context of incorporation during the unique dissolution of cellulose using ionic liquid, nor has it been used in the case of such a regeneration process of cellulose also incorporating filler particles.”

  1. The progress in the application of cellulose in bio-based food packaging films should be briefly supplemented in appropriate location of “Introduction” section.

***ANSWER: We appreciate that this aspect needs such a brief introduction – thank you for the suggestion. We have added:

“Cellulose packaging film has already been introduced as a first step to avoid plastic use, for example, in supermarket coverings and for loose goods [12,13]. Cellulose films are currently existing, but their manufacture so far does not provide potential for the use of circular economy resources. It is this latter aspect that we newly propose as a potential arising from the present work.”

  1. Line 136,the reaction of AKD and cellulose was homogeneous in ionic liquid system, so why was the addition of AKD or hydrophobicity regulated not by the molar number of hydroxyl groups on the AGU but on the mass concentration of hydrophobic agent?

***ANSWER: The reviewer is correct in drawing attention to the functional aspect of the interaction via the anhydroglucose units (AGU), referring to single sugar molecules in a polymer. The behaviour, however, of these hydroxyl groups has not been studied when cellulose is in solution in ionic liquid, which, being a salt melt designed to have both hydrophobic and hydrophilic moieties, could act to block the immediate in-solution interaction with AKD. It is, therefore, at this stage perhaps not wise to speculate as to when the interaction takes place within the complete regeneration process, especially since an aqueous washing step is later applied to remove the ionic liquid, during which the homogeneously distributed AKD might only then undergo the interaction suggested. Considering this and the projected use within an industrial papermaking context, the use of weight fractions is adopted rather than interaction descriptors. We insert the following to provide some supporting discussion:

“It might be questioned as to when the interaction between AKD and cellulose via the hydroxyl groups on the anhydroglucose units (AGU), referring to single sugar molecules in polymer cellulose, actually occurs. The behaviour of these hydroxyl groups has not been studied when cellulose is in solution in ionic liquid, which, being a salt melt designed to have both hydrophobic and hydrophilic moieties, could act to block the immediate in-solution interaction with AKD. We, therefore, at this stage perhaps avoid to speculate as to when the interaction takes place, especially since an aqueous washing step is later applied to remove the ionic liquid, during which the homogeneously distributed AKD might only then undergo the interaction suggested. Considering this and the projected use within an industrial papermaking context, the use of weight fractions is adopted rather than the interaction descriptor AGU.”

  1. For the schematic illustration of the film preparation procedure in Fig. 1(c), pleased indicate which film sample it is.

***ANSWER:  We have added that the ″CA″ films are exemplified in the schematic film preparation procedure in Fig. 1(c), and mentioned this in the caption.

  1. The abrasion resistance property should be evaluated by application properties such as hydrophobicity, oleophobicity, water vapour or oxygen barrier properties, not simply weight loss rate.

***ANSWER: We agree that some further commentary should be made regarding the robustness. Already in response to the reviewers 1 and 2, we shall include contact angle measurement before and after abrasion as well as SEM micrographs before and after abrasion. To report on other properties in detail, however, would require a larger range of abrasion studies, and at this stage of considering laboratory samples only we consider this to be a matter for future work. We add the following text:

“Whilst presenting some limited information on the effect of abrasion, we recognise that the result is only illustrative, being considered here under a very small range of abrasion and environmental conditions, and applied only to hand-made laboratory samples. Therefore, seeking to comment on the whole range of property changes due to abrasion, beyond that of contact angle, such as water vapour transmission, grease resistance etc., we consider to be a matter for future development work.”

  1. Line 298. In FTIR spectra, Fig. 3(a), the adsorption signal at 1420 cm-1 was not obvious. Please present this spectrum so that the peak is visible. You can put this data in the supplementary materials for the article. Additionally, for the sample of CA, why no absorption signals of methyl and methylene groups and ketone ester structure?

***ANSWER: Thank you very much for your insightful comment. We draw attention that those films made from preground copy paper display a slight displacement of the peaks around 1 468, 1 080, 870, and 712 cm−1 confirming the presence of calcite, the most stable polymorph of calcium carbonate, the mineral filler constituent of the paper. In the case of  Fig. 3(a), we have measured the FTIR spectra for the sample of CA separately again, and now absorption signals of methyl and methylene groups and ketone ester structure are clearer. 

  1. Line 338, what are AKD particles?

***ANSWER: The use of the word “particles” here refers to AKD wax-like particles in suspension below the melt temperature of AKD. We enhance the definition in the text as follows:

“The wax-like particles of AKD, below the melt temperature of AKD, …………..”

  1. In Fig. 7(c), starch is hydrophilic, so why the water contact angle of CA-SO+PAO is so much higher than that of CAO+PAO+PAO?

***ANSWER: Starch is used in papermaking as a sizing or hydrophobising agent, this property developing in its dried state. As we also noted earlier, the surface when including starch internally becomes smoother and so the AKD coating is additionally better distributed. Many factors balance the smoothness versus surface energy effects, and at this stage we limited ourselves to reporting the findings as observed. No doubt, further optimisation is needed. We have added the following text:

“Contact angle variation between the various sample formulations relates to the various hydrophobic surface energy components and surface smoothness and distribution of surface roughness present. To elucidate fully on particular values would require a more in-depth analysis of these parameters. We confine ourselves here to seeking the most effective combination to reach highest contact angle.”

  1. Several preparation processes were introduced in the manuscript and their properties were compared and analyzed. Therefore, it is recommended to give the best overall performance example in the summary section.

***ANSWER: This is an attractive request. However, to meet it is not fully possible since various applications of films require different properties, e.g. strength versus plasticity versus elasticity, smoothness versus roughness, hydrophobicity versus hydrophilicity, barrier versus breathable film etc. Nonetheless, to take the attractive target of hydrophobicity we emphasise the difficult to achieve superhydrophicity as some sort of ultimate achievement. We add to the Conclusions as follows:

“Given the many demands on packaging performance ranging from strength versus plasticity versus elasticity, smoothness versus roughness, hydrophobicity versus hydrophilicity, barrier versus breathable film etc., we have set out to illustrate the broad range that the concept described in this work provides. However, to emphasise a single goal, perhaps the ability to reach superhydrophicity together with barrier properties might be singled out as a major technological step.

  1. 3.3 Increasing hydrophilicity to the level of superhydrophobicity, here, “hydrophilicity” should be “hydrophobicity”?

***ANSWER: Well spotted – thank you! We have changed the section title to read “Changing from hydrophilicity to reach the level of superhydrophobicity”

  1. “Silanes, despite their superior performance efficiency compared to wax, are expensive and often considered harmful to the environment”: any proof for this statement?

***ANSWER: Apologies – siloxane was the material that was meant in respect to the environmental discussion. We add the following environmental assessment study references to illustrate the persistence and hazards for siloxanes in the environment:

“Siloxanes, despite their superior performance efficiency when used to hydrophobise paper and board, compared to, say, wax, are expensive and often considered harmful to the environment, particularly in respect to persistence and toxic impact on aquatic systems [20,21,22]”.

  1. English needs to be further polished, especially some sentences are too lengthy for readers: for example:

(1)   “The aim of this study is to investigate if it is possible to incorporate the papermaking sizing agents AKD and starch for the first time into IL dissolved cellulose dopes incorporating calcium carbonate filler particles, to 90 form novel regenerated cellulose benefitting from microparticle strengthening to form composite film barriers.” This sentence is a bit too length, please try to reshape, since it is important for readers to understand the purpose of this work.

***ANSWER: We have both shortened and re-written also the context of this statement in the manuscript.

(2)  By focusing initially on copy paper 98 from office waste, the potential enhancement to establish circular economy constitutes a first for biomass 99 materials not entering into recycling for reasons of incompatibility with common recycling processes.

***ANSWER: Similarly, we shorten the mentioned sentence to read as:

“Furthermore, by focusing initially on copy paper, derived from office waste, we illustrate the potential enhancement to establish circular economy. It also constitutes a first for the continued re-use of biomass materials not entering into recycling for reasons of incompatibility with common recycling processes.”

Submission Date

24 June 2022

Date of this review

11 Jul 2022 12:21:54

Reviewer 4 Report

The author described a kind of regenerated cellulosic composite film with multiple barrier properties such as moisture, water vapor, oxygen and oil resistance, for food packaging applications. The multi-functional cellulose film was prepared by modifying in bulk and surface. AKD is commonly used for hydrophobic modification of cellulose substrate. Although the novelty of this paper is mediocre, the work is relatively comprehensive, and the influence of different preparation processes on the properties of the films was discussed.

1.       The progress in the application of cellulose in bio-based food packaging films should be briefly supplemented in appropriate location of “Introduction” section.

2.       Line 136,the reaction of AKD and cellulose was homogeneous in ionic liquid system, so why was the addition of AKD or hydrophobicity regulated not by the molar number of hydroxyl groups on the AGU but on the mass concentration of hydrophobic agent?

3.       For the schematic illustration of the film preparation procedure in Fig. 1(c), pleased indicate which film sample it is.

4.       The abrasion resistance property should be evaluated by application properties such as hydrophobicity, oleophobicity, water vapour or oxygen barrier properties, not simply weight loss rate.

5.       Line 298. In FTIR spectra, Fig. 3a, the adsorption signal at 1420 cm-1 was not obvious. Please present this spectrum so that the peak is visible. You can put this data in the supplementary materials for the article. Additionally, forthe sample of CA, why no absorption signals of methyl and methylene groups and ketone ester structure?

6.       Line 338, what are AKD particles?

7.       In Fig. 7(c), starch is hydrophilic, so why the water contact angle of CA-SO+PAO is so much higher than that of CAO+PAO+PAO?

8.       Several preparation processes were introduced in the manuscript and their properties were compared and analyzed. Therefore, it is recommended to give the best overall performance example in the summary section.

9.       3.3 Increasing hydrophilicity to the level of superhydrophobicity, here, “hydrophilicity” should be “hydrophobicity”?

10.    “Silanes, despite their superior performance efficiency compared to wax, are expensive and often considered harmful to the environment”: any proof for this statement?

11.    English needs to be further polished, especially some sentences are too lengthy for readers: for example:

(1)   “The aim of this study is to investigate if it is possible to incorporate the papermaking sizing agents AKD and 89 starch for the first time into IL dissolved cellulose dopes incorporating calcium carbonate filler particles, to 90 form novel regenerated cellulose benefitting from microparticle strengthening to form composite film barriers.” This sentence is a bit too length, please try to reshape, since it is important for readers to understand the purpose of this work.

(2)   By focusing initially on copy paper 98 from office waste, the potential enhancement to establish circular economy constitutes a first for biomass 99 materials not entering into recycling for reasons of incompatibility with common recycling processes.

Author Response

Response to Reviewers' Comments:

Ref: sustainability-1809910, "Achieving a superhydrophobic, moisture, oil and gas barrier film using regenerated cellulose-calcium carbonate composite derived from paper components or waste"

We acknowledge the suggestions provided by the reviewers as well as those from the editorial office. All the items are addressed in the revised version of the manuscript. The itemised list below incorporates all comments and point-by-point reply to each item. The corresponding changes are highlighted in the revised manuscript using the “Track Changes” function provided for further processing. We appreciate the suggestions provided by the reviewers and editor, who volunteered insightful comments that definitely helped to enhance the quality and impact of our contribution.

Reviewer 4

Open Review #4

(x) I would not like to sign my review report

( ) I would like to sign my review report

English language and style

(x) Extensive editing of English language and style required

( ) Moderate English changes required

( ) English language and style are fine/minor spell check required

( ) I don't feel qualified to judge about the English language and style

Yes      Can be improved        Must be improved       Not applicable

Is the content succinctly described and contextualized with respect to previous and present theoretical background and empirical research (if applicable) on the topic? ( ) (x)            ( ) ( )   

Are all the cited references relevant to the research? (x)            ( )        ( )        ( )

Are the research design, questions, hypotheses and methods clearly stated? (x) ( ) ( ) ( )

Are the arguments and discussion of findings coherent, balanced and compelling? (x)( )( )     ( )

For empirical research, are the results clearly presented? (x)       ( )        ( )        ( )

Is the article adequately referenced? (x)         ( )        ( )            ( )

Are the conclusions thoroughly supported by the results presented in the article or referenced in secondary literature? (x)  ( )        ( )        ( )

Comments and Suggestions for Authors

The manuscript entitled "Achieving a superhydrophobic, moisture, oil and gas barrier film using regenerated cellulose-calcium carbonate composite derived from paper components or waste" by Imani et al. basically deals with the preparation of cellulose-calcium carbonate composite films from waste/ residues which have a circular economy and ecofriendly approach. However, the manuscript lacks coherence in many places as follows which must be amended for further consideration:

  1. The "Abstract" section is excessively exaggerated with unnecessary detailing! Keep it composed and brief.

***ANSWER: The role of an Abstract is indeed there, as you highlight, to inform concisely and to provide sufficient information for it to stand alone. We have undertaken a total review of the Abstract as follows:

“It has been a persistent challenge to develop eco-friendly packaging cellulose film providing the required multiple barrier properties whilst simultaneously contributing to a circular economy. Typically, a cellulosic film made from nanocellulose materials presents severe limitations, such as poor water/moisture resistance and lacking water vapour barrier properties, related primarily to the hydrophilic and hygroscopic nature of cellulose. In this work, alkyl ketene dimer (AKD) and starch, both eco-friendly, non-toxic, cost-effective materials, were used to achieve barrier properties of novel cellulose-calcium carbonate composite films regenerated from paper components, including paper waste, using ionic liquid as solvent. AKD and starch were applied firstly into the ionic cellulose solution dope mix, and secondly, AKD alone was coated from hot aqueous suspension onto the film surface using a substrate surface precooling technique. The interactions between the AKD and cellulose film were characterised by Fourier transform infrared spectroscopy (FTIR) and X-ray diffraction (XRD) showing the formation of a ketone ester structure between AKD and the hydroxyl groups of cellulose. The presence of calcium carbonate particles in the composite was seen to enhance the cellulose crystallinity. The initial high water vapour and oxygen transmission rates of the untreated base films could be decreased significantly from 2.00 g m-2 s-1 to 0.14 g m-2 s-1, and 3.85×102 cm3 m-2 d-1 to 0.45×102 cm3 m-2 d-1, respectively. In addition, by applying subsequent heat treatment to the AKD coating, the water contact angle was markedly increased to reach levels of superhydrophobicity (>150°, and roll-off angle <5°). The resistance to water absorption, grease-permeation and tensile strength properties were ultimately improved by 41.52%, 95.33%, and 127.33%, respectively, compared with those of an untreated pure cellulose film. The resulting regenerated cellulose-calcium carbonate composite based film and coating formulation can be considered to provide a future bio-based circular economy barrier film, for example, for the packaging, construction and agriculture industries, to complement or replace oil-based plastics.”

  1. The "Introduction" section lacks coherence in different paragraphs! It must be a story overall, which is lacking. The Introduction section also totally lacks the hypotheses and aim of the study! This must be added in the last part of the introduction like any other standard publication. Line no. 88 has unnecessarily many references without much detailing, please take care of that.

***ANSWER: Our response to this need for hypotheses to be embedded in the Introduction has been to insert the important targets at the relevant points in this section. The modified text has been directly inserted into the manuscript.

  1. The methodology totally lacks citations, which must be improved.

***ANSWER: Thank you for raising this point, since it is the core of our activity reported in this manuscript. We consider that the methodologies applied in this novel work are themselves, by definition, novel, and so lack prior art other than that already summarised in the recent publication by Kostic et al. 2022. In fact, (i) the inclusion of filler in such a process has not been reported before this single prior art, (ii) the addition of chemistries, such as that of AKD into IL dissolved cellulose has never been reported before, and, therefore, the methodology to do so is simply reported here so that others can verify our totally novel procedures and findings. It is because of this novelty applied to a target of sustainability that we consider the manuscript an addition of value to the journal Sustainability.

  1. Please provide unedited resolution of optical microscopy images of Fig. 5! Too much colour contrast changes have been made.

***ANSWER: The images of optical microscopy are specifically provided to demonstrate the observation under a variety of highlighting illumination, designed by our institute’s expert microscopist. Rather than lose the benefit of these images in providing the information we discuss, if the editor wishes we can retain them in supplementary information.

  1. The manuscript totally lacks discussion with the relevant publications. Please do compare your result with others in the same field. Also, why the authors are claiming the work is novel is neither clear! Why a separate section "summary" is added is not understandable too. Please revise the manuscript like any other standard scientific manuscript format.

Thank you for highlighting once again the need for us to explain the novelty here. There are no references available to compare IL dissolved cellulose being developed as a barrier film in this way by adding hydrophobising and permeation blocking additives to the IL dissolution process. What we have done is to refer the reader to review data in Figure 8(d), in which existing industrially accepted barrier materials made from oil-based polymers are compared with our result. We show that we are equivalent to PET and PVC for O2 barrier and superior to all reported barrier materials in the review diagram in respect to preventing water vapour transmission. We hope that by drawing the reviewer’s attention to the novelty now explained and the comparison with referenced barriers in use today, that we can convincingly display the merit of our achievement.

***ANSWER: “We recognise there is a difficulty in assessing the degree of success the novel composite material reaches in respect to barrier film properties. This is because there are no references available to compare IL dissolved cellulose being developed as a barrier film in this way by adding hydrophobising and permeation blocking additives to the IL dissolution process. What we do provide, to assist in critical assessment, is to refer the reader to the review data in Figure 8(d), in which existing industrially accepted barrier materials made from oil-based polymers are compared with our result. We show that we have reached values equivalent to PET and PVC for O2 barrier and superior values to all reported barrier materials in the same review in respect to preventing water vapour transmission. We hope that by drawing attention to the novelty and the comparison with referenced barriers in use today, that we can convincingly display the merit of the material achievement.”

  1. The "Conclusion" section is not like a standard "conclusion" section of standard scientific manuscripts. Please conclude the important finding of your result in this section and give only a brief of work to be done (if needed) and future prospects. The summary can be a part of the conclusion instead.

***ANSWER: We have combined the Summary and the Conclusions. This was also explained in the case of Reviewer 2.

Submission Date

24 June 2022

Date of this review

14 Jul 2022 18:52:22

Round 2

Reviewer 1 Report

The revised version of the manuscript seems okay for me, minor spelling checks are required. The hypothetical images may be formed with better resolution and using some scientific image-making tools like biorender to increase the quality of the manuscript.

Reviewer 3 Report

This time, this article is good enough to be published. 

Reviewer 4 Report

The author gave satisfactory answers to the questions raised and the revised manuscript has met the requirements for publication.